# Impaired lipid biosynthesis hinders anti-tumor efficacy of intratumoral iNKT cells

Sicheng Fu[1,2], Kaixin He[2], Chenxi Tian[1,2], Hua Sun[3], Chenwen Zhu[3], Shiyu Bai[1,2], Jiwei Liu[1,2], Qielan Wu[1,2], Di Xie[1,2], Ting Yue[2], Zhuxia Shen[4], Qingqing Dai[5], Xiaojun Yu[5], Shu Zhu [2], Gang Liu[6], Rongbin Zhou [2], Shengzhong Duan[7,8], Zhigang Tian [1,2], Tao Xu [9], Hua Wang [3]* & Li Bai [1,2]*

Dysfunction of invariant natural killer T (iNKT) cells in tumor microenvironment hinders their anti-tumor efficacy, and the underlying mechanisms remain unclear. Here we report that iNKT cells increase lipid biosynthesis after activation, and that is promoted by PPARγ and PLZF synergically through enhancing transcription of *Srebf1*. Among those lipids, cholesterol is required for the optimal IFN-γ production from iNKT cells. Lactic acid in tumor micro-environment reduces expression of PPARγ in intratumoral iNKT cells and consequently diminishes their cholesterol synthesis and IFN-γ production. Importantly, PPARγ agonist pioglitazone, a thiazolidinedione drug for type 2 diabetes, successfully restores IFN-γ production in tumor-infiltrating iNKT cells from both human patients and mouse models. Combination of pioglitazone and alpha-galactosylceramide treatments significantly enhances iNKT cell-mediated anti-tumor immune responses and prolongs survival of tumor-bearing mice. Our studies provide a strategy to augment the anti-tumor efficacy of iNKT cell-based immunotherapies via promoting their lipid biosynthesis.

[1] Department of Oncology of The First Affiliated Hospital, the CAS Key Laboratory of Innate Immunity and Chronic Disease, Division of Life Sciences and Medicine, University of Science and Technology of China, Hefei 230027, China. [2] Hefei National Laboratory for Physical Sciences at Microscale, School of Life Sciences, Division of Life Sciences and Medicine, University of Science and Technology of China, Hefei 230027, China. [3] Department of Oncology, The First Affiliated Hospital of Anhui Medical University, Anhui Medical University, Hefei 230022, China. [4] Department of Cardiology, Jing'an District Centre Hospital of Shanghai, Fudan University, Shanghai 200040, China. [5] Department of Hepatopancreatobiliary Surgery and Organ Transplantation Center, Department of General Surgery, The First Affiliated Hospital of Anhui Medical University, Hefei 230022, China. [6] National Synchrotron Radiation Laboratory, University of Science and Technology of China, Hefei 230027, China. [7] Laboratory of Oral Microbiota and Systemic Diseases, Shanghai Ninth People's Hospital, College of Stomatology, Shanghai Jiao Tong University School of Medicine, Shanghai 200125, China. [8] National Clinical Research Center for Oral Diseases, Shanghai Key Laboratory of Stomatology and Shanghai Research Institute of Stomatology, Shanghai 200011, China. [9] National Key Laboratory of Biomacromolecules, Institute of Biophysics, Chinese Academy of Sciences, Beijing 100101, China. *email: wanghua@ahmu.edu.cn; baili@ustc.edu.cn

mmunotherapies have been suggested as the most specific and effective treatments against tumors. Except for the cytotoxic T lymphocytes (CTLs) and natural killer (NK) cells, invariant natural killer T (iNKT) cells play important roles in anti-tumor immune responses as well. They could either kill tumor cells directly through antigen recognition or augment anti-tumor responses by depleting tumor associated macrophages (TAMs) and promoting activation of NK and CTLs[1–4]. Therefore, iNKT cells are promising candidates for anti-tumor immunotherapies. Administration of alpha-galactosylceramide (αGC), a ligand for iNKT cells, or αGC-pulsed DCs inhibits tumor growth in both mouse models and human patients[5–8]. Moreover, recent studies report a notable anti-tumor efficacy of CAR iNKT cells[9,10]. However, not all patients respond to these iNKT cell-based immunotherapies[7,8]. Those failures are linked to impaired iNKT cell function. Although previous studies suggest that TAMs[11–13], myeloid derived suppressor cells[14], regulatory T cells[15,16], exosomes[17], metabolic products[18,19], and suppressive cytokines[20] in tumors repress anti-tumor immunity and cause the failure of immunotherapies, the mechanisms underlying dysfunction of intratumoral iNKT cells still remain unclear. Elucidating the underlying mechanisms would help to design new immunotherapies augmenting the efficacy of iNKT cell-based immunotherapies.

Recent studies demonstrate a link between cellular metabolism and immune cell functions. Impaired metabolism has been reported as a main reason causing the dysfunction of anti-tumor immune cells in tumor microenvironment[21,22]. Whether metabolism regulates function of iNKT cells in tumors remains unclear. Here, we show that activation of iNKT cells increases their lipid biosynthesis, and that is promoted by PPARγ in cooperation with PLZF and is required for the optimal IFN-γ production. Lactic acid in tumor microenvironment reduces PPARγ expression in intratumoral iNKT cells and thereby diminishes their lipid synthesis and IFN-γ production. Importantly, activation of PPARγ via pioglitazone (PIO), a drug used to treat type 2 diabetes, restores lipid synthesis and IFN-γ production in those intratumoral iNKT cells, and significantly enhances anti-tumor efficacy of iNKT cell-based immunotherapy. Our results demonstrate the mechanisms underlying dysfunction of iNKT cells in tumor microenvironment, and provide important insights into interventions enhancing iNKT cell-mediated anti-tumor immune responses.

## Results

**PPARγ promotes lipid biosynthesis in iNKT cells**. As the major component of cells, lipids have been reported to regulate the differentiation and function of immune cells. However, their roles in iNKT cells remain unclear. We detected higher level of PPARγ (Fig. 1a), a well-known regulator of adipocyte differentiation and lipid metabolism, in sorted mouse iNKT cells (Supplementary Fig. 1) in comparison to conventional CD8+ and CD4+ T cells. The higher expression of PPARγ in iNKT cells was further confirmed at protein level by flow cytometry and at mRNA level by qPCR (Fig. 1b, c). Notably, anti-CD3 plus anti-CD28 stimulation further increased PPARγ expression (Fig. 1d), as well as the lipid amount in iNKT cells (Fig. 1e) as indicated by the staining of 4,4-difluoro-1,3,5,7,8-pentamethyl-4-bora-3a,4a-diaza-s-indacene (BODIPY). Next, we analyzed fatty acid composition in iNKT cells, and found that anti-CD3 plus anti-CD28 stimulation significantly increased amount of intracellular fatty acids, including C14:0, C16:0, C16:1, C18:1n9t, and C18:2n6t (Fig. 1f). PIO, the agonist of PPARγ, further enhanced the amount of these intracellular fatty acids in activated iNKT cells, whereas T007, the antagonist of PPARγ, significantly reduced the amount of

these fatty acids (Fig. 1f). These results suggested that PPARγ activity controlled the amount of fatty acids in iNKT cells. In addition, we found that mRNA of genes regulating fatty acid biosynthesis were increased in iNKT cells after activation, and these mRNA levels were further enhanced by PIO and reduced by T007 (Fig. 1g). All these data demonstrate that PPARγ promotes fatty acid biosynthesis in iNKT cells after activation.

**PPARγ-controlled lipid synthesis promotes IFN-γ production**. Next, we investigated the functional impacts of PPARγ-controlled lipid synthesis in iNKT cells. We activated iNKT cells with either plate-coated anti-CD3 plus anti-CD28 antibodies or with plate-coated mCD1d-PBS57 tetramers, and inhibited lipid synthesis with antagonist of PPARγ, T007, or with inhibitor of fatty acid synthesis, Tofa. After activation, iNKT cells proliferated and increased expression of activation markers CD69 and CD25. T007 and Tofa significantly reduced expression of CD69 and CD25 (Fig. 2a, b), and inhibited cell proliferation (Fig. 2c). Interestingly, T007 and Tofa dramatically inhibited IFN-γ production from sorted iNKT cells in a dose dependent manner, whereas showed only minor effect on their IL-4 production (Fig. 2d, e). These results indicated polarization of iNKT cell functions toward Th2 response. In addition to IFN-γ, reduced Th1 cytokines IL-2 and TNF-α were detected in iNKT cells treated with GW9662, another antagonist of PPARγ (Supplementary Fig. 2a, b). In agreement with its in vitro effects, GW9662 also inhibited αGC induced IFN-γ production in vivo. Frequency of IFN-γ+ hepatic iNKT cells and mean fluorescence intensity of intracellular anti-IFN-γ were reduced by GW9662, whereas no effect of GW9662 on IL-4 production was detected (Supplementary Fig. 2c, d). Moreover, we found that *Ifng* mRNA was dramatically reduced in iNKT cells treated with PPARγ antagonists, including GW9662 and T007, or with inhibitors of fatty acids synthesis, including Tofa and C75 (Fig. 2f, g). Again, these inhibitors showed minor effects on *Il4* mRNA level (Fig. 2f, g). These results indicated that PPARγ-controlled lipid synthesis promoted IFN-γ production in iNKT cells at the transcriptional level. To further confirm the role of PPARγ in iNKT cells, we used shRNA to knock down its expression (Fig. 2h). Knockdown of PPARγ significantly reduced IFN-γ production (Fig. 2i). In addition, by crossing *Pparg^fl/fl* mice with PLZF-cre mice, we deleted PPARγ in iNKT cells but not in conventional T cells (Fig. 2j). PPARγ deficiency reduced iNKT cell frequencies in thymuses but not in spleens or livers from PLZF-cre *Pparg^fl/fl* mice (Supplementary Fig. 3). In line with the knockdown of PPARγ, deletion of PPARγ in iNKT cells reduced their IFN-γ production when cells were activated in vitro (Fig. 2k). Moreover, we showed that PIO increased IFN-γ production and T007 reduced IFN-γ production in wide type iNKT cells but not in PPARγ deficient iNKT cells (Fig. 2k). These results further confirmed that PIO and T007 regulated IFN-γ production in iNKT cells by targeting PPARγ. Taken together, our results demonstrate that PPARγ promotes activation and IFN-γ production in iNKT cells via enhancing lipid synthesis.

Although fatty acid β-oxidation has been reported to regulate T cell fate[23,24], transcription of *Cpt1a*, a gene controlling fatty acid β-oxidation, was reduced after activating iNKT cells. Inhibitor of CPT1α, Etomoxir, showed no influence on IFN-γ or IL-4 production from iNKT cells (Supplementary Fig. 4). Therefore, fatty acid β-oxidation is not required for iNKT cell effector functions.

**Cholesterol is required for IFN-γ production in iNKT cells**. Previous findings have shown that certain of saturated fatty acids and unsaturated fatty acids play important roles in early

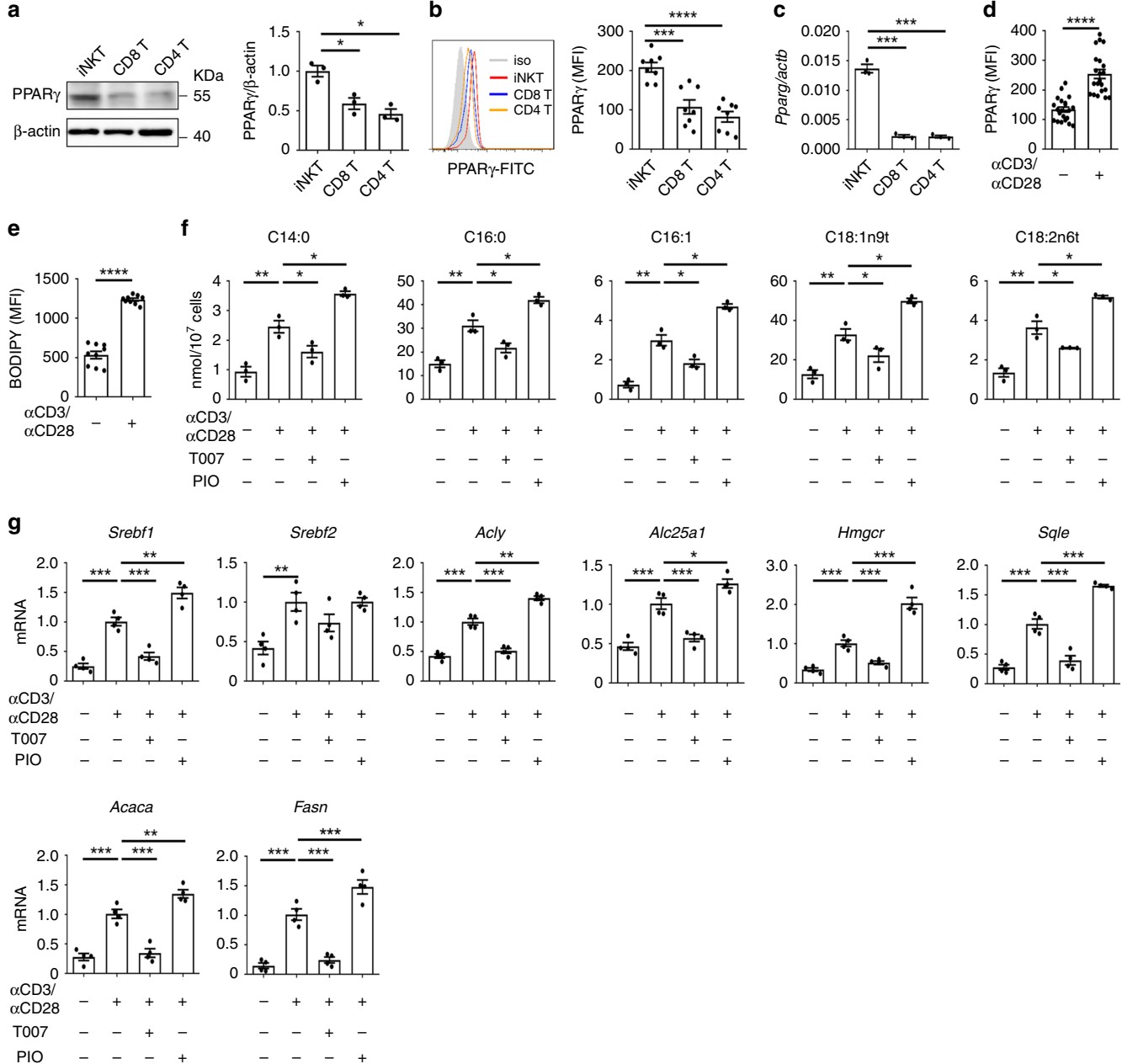

**Fig. 1 PPARγ promotes lipid biosynthesis in activated iNKT cells. a** Expression of PPARγ in iNKT, CD4+ T, and CD8+ T cells. **b**, **c** PPARγ expression measured by flow cytometry (**b**) and mRNA of *Pparg* (**c**) in iNKT, CD4+ T, and CD8+ T cells from livers. **d**, **e** Flow cytometry analysis of PPARγ expression (**d**) and lipids amount indicated by BODIPY staining (**e**) in iNKT cells unstimulated or stimulated with plate-coated anti-CD3 and anti-CD28 overnight. **f** Quantification of long-chain fatty acids in iNKT cells, 24 h after activation in vitro with or without T007, PIO. **g** mRNA of genes regulating lipid synthesis in iNKT cells activated by plate-coated anti-CD3 and anti-CD28 for 24 h with or without T007, PIO. Data are representative of three independent experiments (**a**, **b**), or are means ± SEM of three independent experiments (**a**, **c**, **f**), four independent experiments (**g**), 8 mice (**b**), nine biological replicates (**e**), or twenty biological replicates (**d**), pooled from three to four independent experiments. Data were analyzed by Mann–Whitney test (**a–c**, **f**, **g**) or unpaired Student's *t*-test (**d**, **e**). *$P < 0.05$, **$P < 0.01$, ****$P < 0.0001$. Source data are provided as a Source Data file.

activation of T cells[25,26]. It is possible that diminished IFN-γ production caused by those inhibitors results from shortage of certain lipid. To identify the lipid controlling IFN-γ production in iNKT cells, distinct fatty acids were, respectively, added to iNKT cells treated with Tofa. Both saturated and unsaturated fatty acids, including octanoic acid, decanoic acid, palmitic acid, oleic acid, linolenic acid, and linoleic acid, failed to restore IFN-γ production (Fig. 3a). High concentration of palmitic acid, oleic acid, and linoleic acid even reduced IFN-γ production (Fig. 3a), possibly due to the increased cell apoptosis[27]. In contrast,

supplementation of water-soluble cholesterol successfully restored IFN-γ production (Fig. 3b). In addition, cholesterol also restored IFN-γ production in iNKT cells treated with T007 (Fig. 3c). The recovery of IFN-γ by cholesterol was in line with the findings that T007 and Tofa reduced total cholesterol, as well as plasma membrane cholesterol in iNKT cells (Fig. 3d). These results indicated that inhibition of lipid synthesis diminished IFN-γ production by reducing cholesterol. Moreover, we showed that simvastatin (Sim), an inhibitor for cholesterol synthesis, significantly inhibited IFN-γ production in iNKT cells (Fig. 3e),

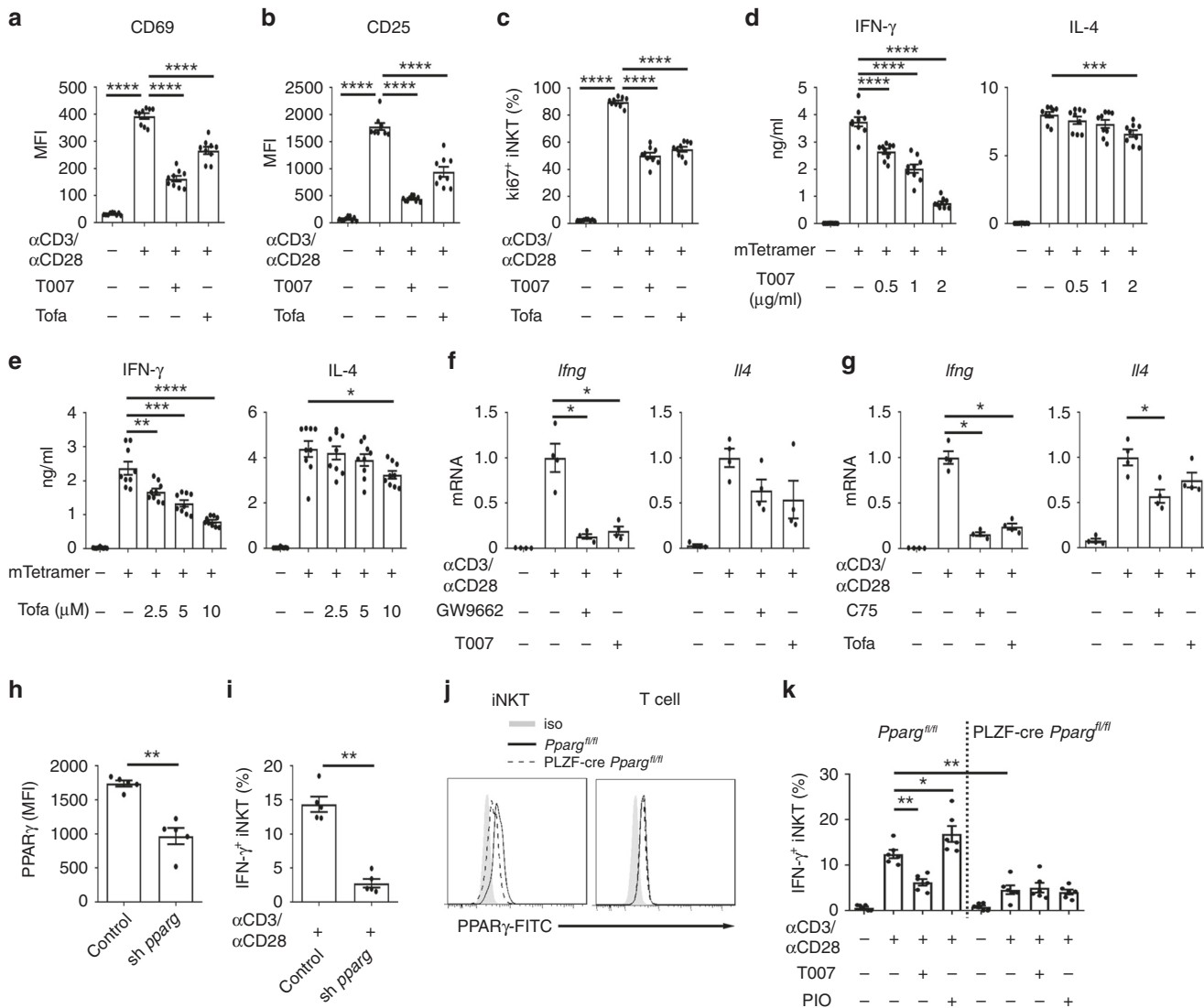

**Fig. 2 PPARγ and lipid synthesis promote activation and IFN-γ production of iNKT cells. a, b** Surface CD69 (**a**), CD25 (**b**) on iNKT cells after activating by plate-coated anti-CD3 and anti-CD28 in the absence or presence of T007, Tofa. Unstimulated iNKT cells were used as negative controls. **c** Frequencies of Ki67+ iNKT cells after activating with plate-coated anti-CD3 and anti-CD28 for 2 days with or without T007, Tofa. **d** IFN-γ and IL-4 production in iNKT cells activated by plate-coated mCD1d-PBS57 tetramer in the absence or presence of T007. **e** IFN-γ and IL-4 production in iNKT cells in the absence or presence of Tofa as described in **d**. **f, g** mRNA of *Ifng* and *Il4* in iNKT cells activated by anti-CD3 plus anti-CD28 for 24 h with or without antagonists of PPARγ (**f**) or fatty acid synthesis inhibitors (**g**). **h, i** Knockdown efficiency of *Pparg* shRNA (**h**) and its effect on percentages of IFN-γ+ iNKT cells, after activating with plate-coated anti-CD3 and anti-CD28 (**i**). **j** PPARγ expression in iNKT cells or T cells from PLZF-cre *Pparg*fl/fl mice or *Pparg*fl/fl mice. **k** Percentages of IFN-γ+ iNKT cells from PLZF-cre *Pparg*fl/fl mice or *Pparg*fl/fl mice, after activating with plate-coated anti-CD3 and anti-CD28 with or without T007, or PIO. Data are representative of six mice (**j**), or are means ± SEM of three independent experiments (**h, i**), nine biological replicates (**a-e**), four independent experiments (**f, g**), or six mice (**k**), pooled from three to four independent experiments. Data were analyzed by unpaired Student's *t*-test (**a-e**) or Mann-Whitney test (**f-i, k**). *P < 0.05, **P < 0.01, ***P < 0.001, ****P < 0.0001. Source data are provided as a Source Data file.

whereas addition of cholesterol elevated their IFN-γ production (Fig. 3f). Again, the amount of cholesterol only slightly influenced IL-4 production (Fig. 3e, f). In line with the role of lipid synthesis in promoting iNKT cell activation and proliferation, we found that Sim inhibited upregulation of CD69 and iNKT cell proliferation (Supplementary Fig. 5a, b). Taken together, these results demonstrate that reduction of cholesterol is responsible for the suppressive effect of impaired lipid synthesis.

Cholesterol has been previously shown to regulate the TCR signaling[28–30]. In our studies, when iNKT cells were activated by PMA plus ionomycin bypassing TCR signaling, exogenous cholesterol, C75, and Tofa showed no influence on IFN-γ production (Fig. 3g). Therefore, cholesterol regulated IFN-γ

production in iNKT cells likely by modulating proximal TCR signaling. Clustering and accumulation of TCR at immunological synapse were inhibited by T007, Tofa, and Sim, as indicated by decreased fluorescence intensity of TCR (Fig. 3h, i). The reduced area of immune synapses indicated impaired formation of synapse (Fig. 3j). To further investigate the activation of TCR signaling pathways by western blot, we used expanded iNKT cells which responded similarly to T007 as fresh iNKT cells did (Supplementary Fig. 6). When expanded iNKT cells were activated and treated with T007, Tofa, and Sim, respectively, decreased phosphorylation of Lck and LAT were detected, confirming impaired proximal TCR signaling (Fig. 3k). Again, supplementation of water-soluble cholesterol successfully restored

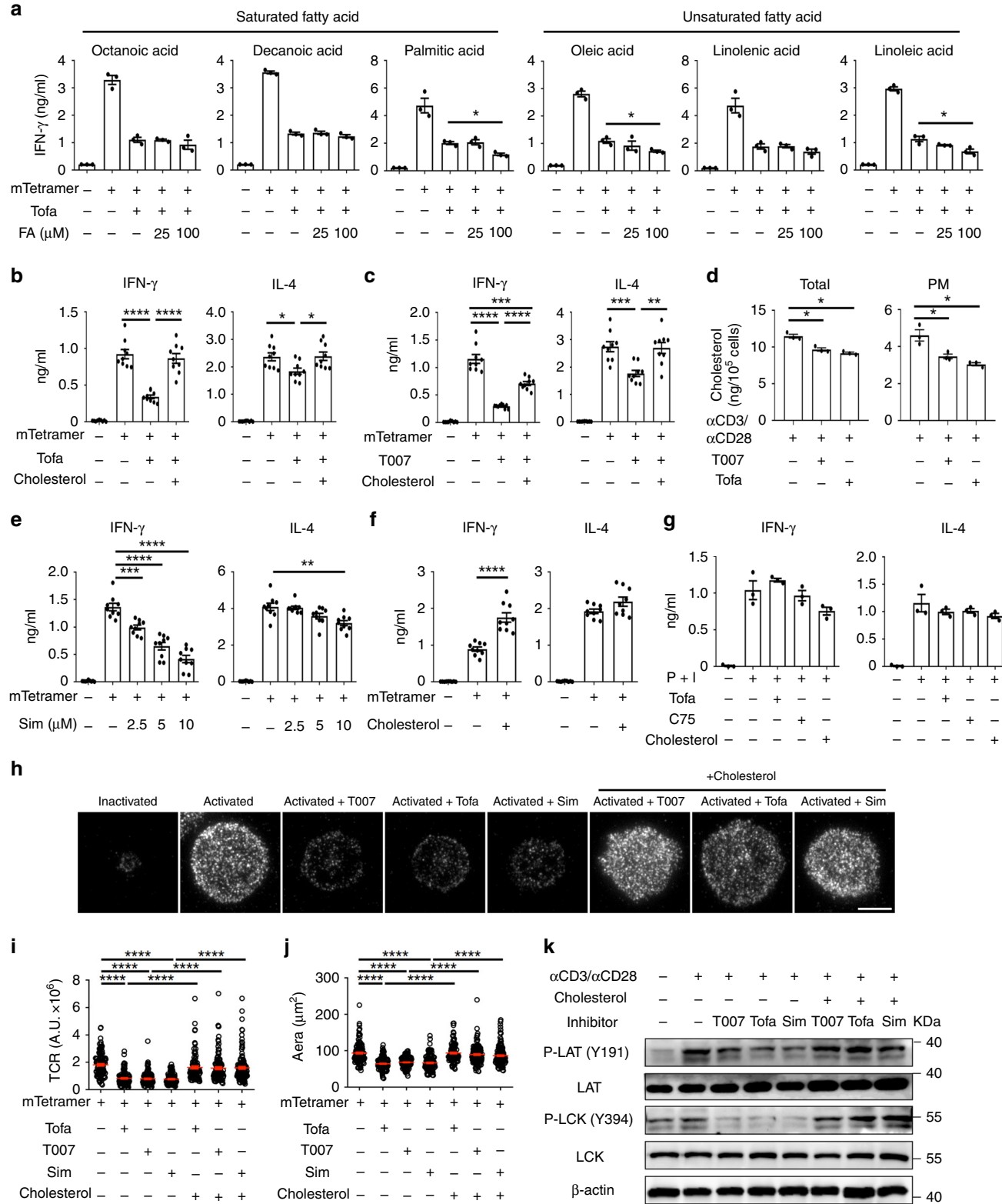

synaptic TCR accumulation, immune synapse formation, and phosphorylation of Lck and LAT (Fig. 3h-k). These results demonstrate that cholesterol enhances TCR signaling at immunological synapse of iNKT cells.

**PPARγ2 and PLZF synergically promotes *Srebf1* transcription.** PPARγ has been previously shown to promote fatty acid uptake in CD4[+] T cells[25]. However, antagonists of PPARγ reduced genes

controlling cholesterol synthesis, including *Srebf1*, *Hmgcr*, and *Sqle* (Fig. 1g), but showed no influence on genes controlling cholesterol efflux or uptake, including *Abca1*, *Abcg1*, *Ldlr*, and *Ldol* (Supplementary Fig. 7). Among those genes controlled by PPARγ, *Srebf1* encodes sterol regulatory element-binding protein 1 (SREBP1), a major transcription factor regulating the biosynthesis of lipids[31]. In agreement with the amount of *Srebf1* mRNA (Fig. 1g), SREBP1 protein level was increased after cell

**Fig. 3 Cholesterol restores IFNγ production in iNKT cells lacking lipid synthesis. a** Influences of indicated fatty acids on IFN-γ production in iNKT cells activated by plate-coated mCD1d-PBS57 tetramer in the presence of Tofa. **b** Influences of water-soluble cholesterol on IFN-γ and IL-4 production from iNKT cells activated by plate-coated mCD1d-PBS57 tetramer in the presence of Tofa. **c** Influences of water-soluble cholesterol on IFN-γ and IL-4 production in iNKT cells activated by plate-coated mCD1d-PBS57 tetramer in the presence of T007. **d** Total cholesterol or plasma membrane cholesterol in iNKT cells activated by anti-CD3 plus anti-CD28 for 24 h with or without T007, or Tofa. **e** Influences of Sim on IFN-γ and IL-4 production in iNKT cells activated by mCD1d-PBS57 tetramer. **f** Influences of water-soluble cholesterol on IFN-γ and IL-4 production in iNKT cells activated by mCD1d-PBS57 tetramer. **g** Influences of Tofa, C75, and cholesterol on IFN-γ and IL-4 production from iNKT cells activated by PMA plus ionomycin. **h–j** Distribution (**h**) and fluorescence intensity (**i**) of surface TCR, and area of immunological synapse (**j**) in iNKT cells activated by coverslip-coated mCD1d-PBS57 tetramer for 30 min in the presence or absence of T007, Tofa, Sim, or water-soluble cholesterol ($n = 100$ cells per group). Arbitary unit, A.U. Bar, 5 μm. **k** P-LCK (Y394), LCK, P-LAT (Y191), LAT in iNKT cells activated by plate-coated anti-CD3 plus anti-CD28 for 20 min in the presence or absence of T007, Tofa, Sim, or water-soluble cholesterol. Data are means ± SEM of three independent experiments (**a**, **d**, **g**) or nine biological replicates (**b**, **c**, **e**, **f**), pooled from three independent experiments, or are representative of three independent experiments (**h**, **k**). Data were analyzed by Mann–Whitney test (**a**, **d**, **g**) or unpaired Student's t-test (**b**, **c**, **e**, **f**, **i**, **j**). *$P < 0.05$; **$P < 0.01$; ***$P < 0.001$, ****$P < 0.0001$. Source data are provided as a Source Data file.

activation and was reduced by T007, in both mature and immature forms (Fig. 4a). The reduction of total SREBP1 protein in presence of T007 was further confirmed by flow cytometry analysis (Fig. 4b). On the other hand, both *Srebf2* mRNA level (Fig. 1g) and SREBP2 protein level were not influenced by T007 (Fig. 4a). Knockdown of PPARγ significantly reduced expression of SREBP1 in activated iNKT cells (Fig. 4c), confirming that PPARγ promoted expression of SREBP1 in iNKT cells after activation. To demonstrate that reduction of SREBP1 could inhibit cholesterol synthesis in iNKT cells, we knocked down SREBP1 by shRNA (Fig. 4c) and measured the cholesterol level by Filipin III staining. Knockdown of SREBP1 decreased cholesterol level as efficiently as knockdown of PPARγ (Fig. 4d). In agreement with the roles of cholesterol in promoting IFN-γγ production, knockdown of SREBP1 inhibited IFN-γ production as well (Fig. 4e). Consistently, we found reduced SREBP1 and cholesterol level in PPARγ deficient iNKT cells (Fig. 4f, g). Together, these results suggest that PPARγ enhances cholesterol synthesis and IFN-γ production in iNKT cells via promoting expression of SREBP1.

To investigate whether PPARγ could directly regulate transcription of *Srebf1*, PPARγ-binding DNA fragments from iNKT cells were collected by chromatin immunoprecipitation, and different sets of primers covering the −2000 bp to +1 bp region of *Srebf1* promoter were used to scan the potential binding sites of PPARγ (Fig. 4h). A pair of primers targeting the known PPAR response element (PPRE) at *Nr1h3* promoter region[32] was used as the positive control. Increased binding of PPARγ to fragment P3 (−359 to −477 bp), P4 (−613 to −730 bp), and P7 (−1338 to −1445 bp) were detected in activated iNKT cells (Fig. 4i). Moreover, we found that T007 completely inhibited binding of PPARγ to these fragments (Fig. 4i). These results indicate direct binding of PPARγ at promoter region of *Srebf1* in activated iNKT cells.

Notably, antagonists of PPARγ did not influence the amount of mature or immature forms of SREBP1 or total SREBP1 protein in CD4$^+$ T and CD8$^+$ T cells after activation (Supplementary Fig. 8a-d). Consistently, binding of PPARγ to *Srebf1* promoter was not influenced by the activation status or by PPARγ antagonist in conventional T cells (Supplementary Fig. 8e, f). As a result, T007 and GW9662 showed no influence on cholesterol synthesis in these conventional T cells (Supplementary Fig. 8g, h). These results indicate that PPARγ regulates expression of SREBP1 in a cell type specific manner. PLZF is a cell type-specific master regulator that controls development and effector functions of iNKT cells, and is not expressed by conventional T cells[33]. We detected PLZF among the proteins co-immunoprecipitated with PPARγ. Activation of iNKT cells further promoted the binding of PPARγ to PLZF, though antagonist and agonist of PPARγ did not influence the

association of these two proteins (Fig. 4j). Moreover, increased co-localization between PLZF and PPARγ was observed after activating iNKT cells (Fig. 4k, l). Again, PIO and T007 showed no influence on the co-localization coefficient. These results indicate that interaction between PPARγ and PLZF is controlled by activation signal other than ligands of PPARγ. Importantly, the interaction between PPARγ and PLZF implied a possible explanation for the cell type-specific regulatory role of PPARγ. Next, we investigated whether PLZF was required for PPARγ to promote *Srebf1* transcription. There are two isoforms of PPARγ, PPARγ1, and PPARγ2. We detected both *Pparg1* and *Pparg2* mRNA in iNKT cells (Fig. 4m). The antibody against PPARγ used in our studies was unable to distinguish PPARγ1 from PPARγ2. PPARγ and PLZF are conserved between mouse and human (96% and 97% homology). Therefore, we overexpressed mouse PPARγ1 and PPARγ2, respectively, in human cell line 293T cells, with or without co-overexpressing mouse PLZF. Overexpressing PLZF, PPARγ1, or PPARγ2 alone failed to promote the transcription and expression of SREBP1 in 293T cells. Co-overexpression of PPARγ1 and PLZF also failed to do so. Interestingly, co-overexpressing PPARγ2 and PLZF significantly increased amount of mature SREBP1, total SREBP1 protein, and *Srebf1* mRNA (Fig. 4n-p). These results demonstrated that PPARγ2 cooperated with PLZF to promote the transcription and expression of SREBP1. Consistently, co-overexpressing PPARγ2 and PLZF in iNKT cells increased SREBP1 expression as well, whereas co-overexpressing PPARγ1 and PLZF did not (Fig. 4q). Comparing to PPARγ1, PPARγ2 has additional 30 amino acids at N-terminal. Co-immunoprecipitation experiments demonstrated that overexpressed PPARγ2 rather than PPARγ1 bound PLZF in 293T cells (Fig. 4r). Therefore, the additional 30 amino acids in PPARγ2 were important for its interaction with PLZF. These results also explained the findings that some PPARγ protein, likely PPARγ1, was not co-localized with PLZF in iNKT cells (Fig. 4k), and PPARγ2 but not PPARγ1 cooperated with PLZF in promoting *Srebf1* transcription (Fig. 4n-q). Furthermore, we found that co-expression of PLZF significantly enhanced binding of PPARγ2 to the *Srebf1* promoter region in 293T cells (Fig. 4s). Together, our results demonstrate that PLZF is required for PPARγ2 to bind at the promoter region of *Srebf1* and to promote its transcription.

In line with our conclusion, iNKT2 and iNKT17 subsets that expressed higher amount of PLZF and PPARγ showed higher amount of SREBP1 and cholesterol in comparison to iNKT1 subset (Supplementary Fig. 9b, c). Similar to conventional T cells, adipose tissue iNKT cells, which expressed PPARγ but negligible PLZF (Supplementary Fig. 9d)[34], did not change their amount of SREBP1 or cholesterol in response to PIO or T007 treatment (Supplementary Fig. 9e).

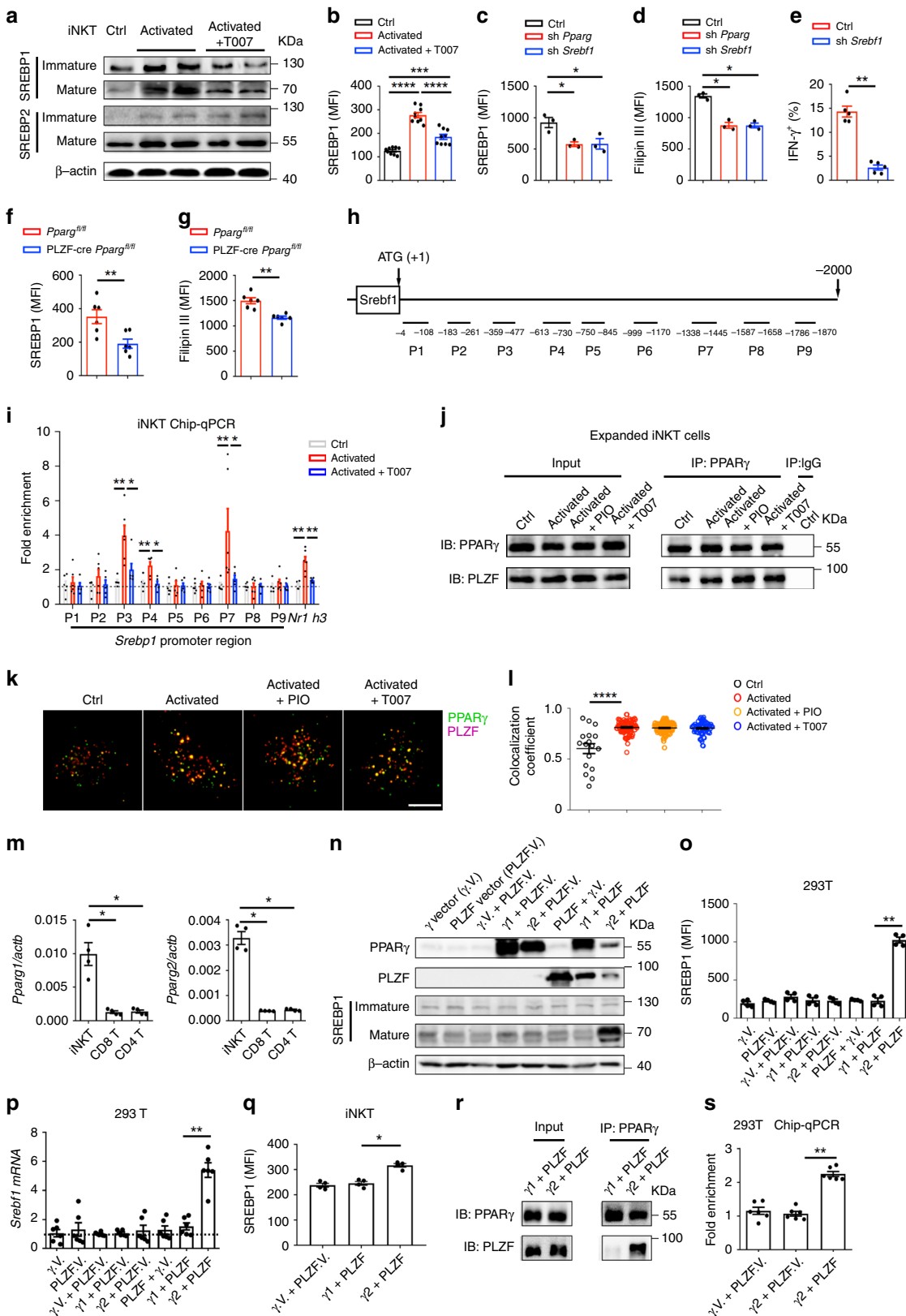

**Reduced PPARγ and IFN-γ in human intratumoral iNKT cells**. Previous findings indicate that altered cellular metabolism influences immune cell functions in tumor microenvironment. The metabolic state and effector functions of iNKT cells in tumor microenvironment remain unclear. In hepatocellular carcinoma (HCC) patients, reduced PPARγ was detected in human tumor-infiltrating iNKT cells, in comparison to cells in para-carcinoma tissues (Fig. 5a). In agreement with our findings that PPARγ promoted expression of SREBP1 and synthesis of cholesterol in mouse iNKT cells (Fig. 4a-d, f, g), reduced SREBP1 (Fig. 5b) and cholesterol levels (Fig. 5c) were detected in human tumor-infiltrating iNKT cells, and the level of SREBP1 and cholesterol

**Fig. 4 PPARγ directly promotes transcription of *Srebf1* in cooperation with PLZF. a** Influences of T007 on mature and immature forms of SREBP1 and SREBP2 in expanded iNKT cells after activating overnight. **b** Total SREBP1 protein in iNKT cells measured by flow cytometry. **c–e** Effects of *Pparg* shRNA or *Srebf1* shRNA on SREBP1 expression (**c**), Fillipin III staining (**d**), and percentages of IFN-γ⁺ iNKT cells after activation (**e**). **f, g** SREBP1 expression (**f**) and Fillipin III staining (**g**) in iNKT from PLZF-cre *Pparg^fl/fl* mice and *Pparg^fl/fl* mice. **h, i** Fragments in *Srebf1* promoter region (**h**) and binding of these fragments to PPARγ measured by ChIP-qPCR (**i**). **j** Co-immunoprecipitation of PPARγ and PLZF in expanded iNKT cells with indicated treatments. (**k, l**) Co-localization (**k**) and co-localization coefficient (**l**) of PPARγ and PLZF in iNKT cells with indicated treatments. Bar, 5 μm. **m** mRNA of *Pparg1* and *Pparg2* in iNKT, CD4⁺ T, and CD8⁺ T cells. **n** Expression of PPARγ, PLZF, and SREBP1 in 293 T cells overexpressing PPARγ1, PPARγ2, PLZF, PPARγ vector (γ.V.), and PLZF vector (PLZF.V.) alone or in combination. **o, p** Total SREBP1 protein measured by flow cytometry (**o**) and mRNA of *Srebf1* (**p**) in 293 T cells described in **n**. (**q**) Total SREBP1 protein in iNKT cells co-overexpressing PPARγ1 and PLZF, or co-overexpressing PPARγ2 and PLZF. **r** Co-immunoprecipitation of PPARγ and PLZF in 293 T cells co-overexpressing PPARγ1 and PLZF, or co-overexpressing PPARγ2 and PLZF. **s** Binding of PPARγ to putative PPRE (ATCTCACAGGTCA) at *Srebf1* promoter region in 293T cells overexpressing PPARγ2 and PLZF as indicated. Data are representative of three independent experiments (**a, j, k, n, r**), or are means ± SEM of three (**c, d**) or four independent experiments (**m, o, q**), nine (**b**) or six biological replicates (**i, p, s**), or six mice (**f, g**), pooled from three independent experiments. Data were analyzed by unpaired Student's *t*-test (**b, l**) or Mann–Whitney test (**c–g, i, m, o, p, q, s**). *P < 0.05, **P < 0.01, ***P < 0.001, ****P < 0.0001. Source data are provided as a Source Data file.

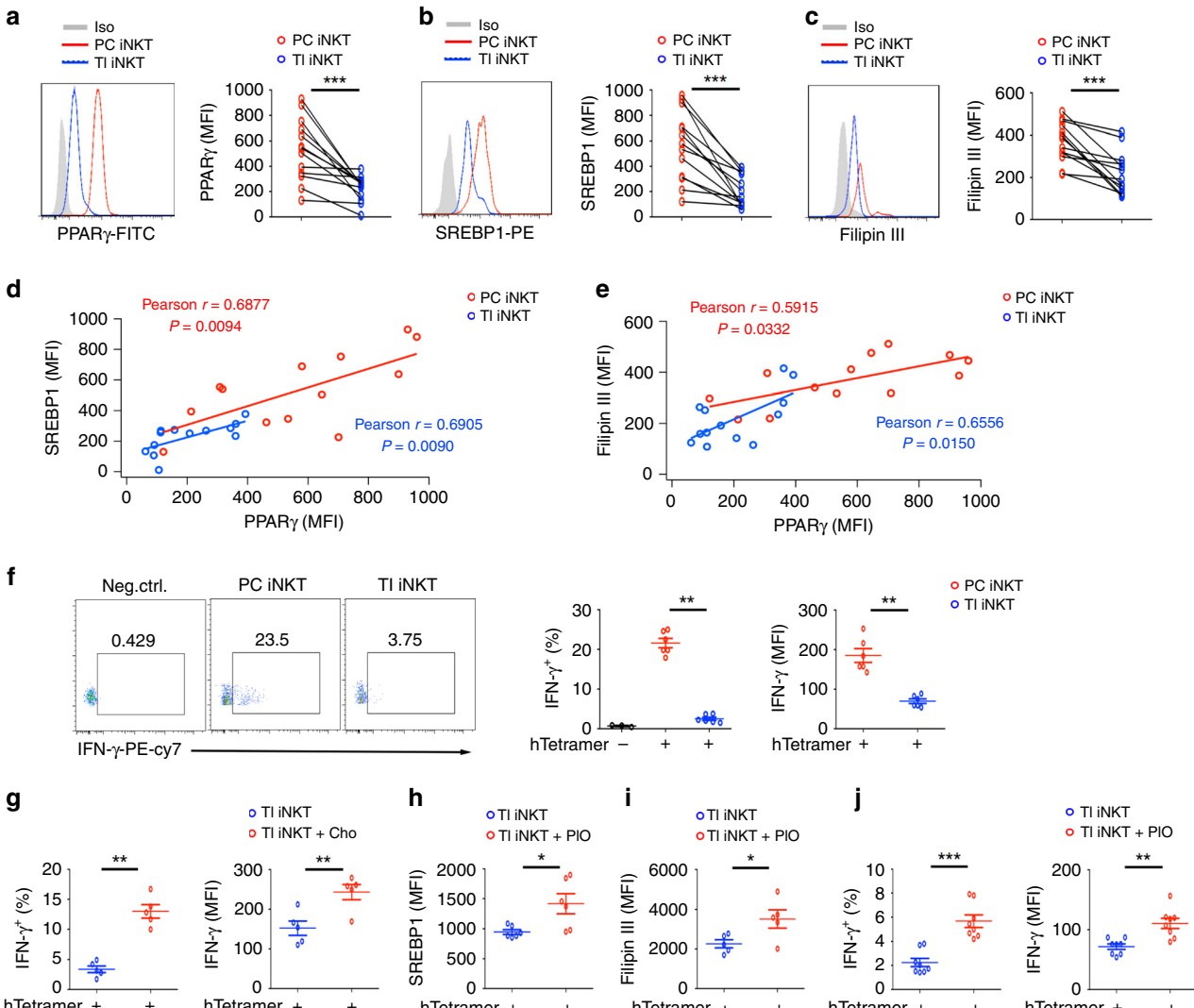

**Fig. 5 PIO restores IFN-γ production in human tumor-infiltrating iNKT cells in vitro. a–c** PPARγ (**a**), SREBP1 (**b**), and Fillipin III staining (**c**) in tumor-infiltrating (TI) and para-carcinoma (PC) iNKT cells from HCC patients. **d, e** Correlations between PPARγ and SREBP1 (**d**), PPARγ and Fillipin III (**e**), in TI and PC iNKT cells as described in **a**. **f** Intracellular IFN-γ staining (left), percentages of IFN-γ⁺ iNKT cells (middle), and mean fluorescence intensity (MFI) of anti-IFN-γ (right) in TI and PC iNKT cells from HCC patients. **g** Influences of water-soluble cholesterol on percentages of IFN-γ⁺ iNKT cells and MFI of anti-IFN-γ in TI iNKT cells from HCC patients. **h–j** Influences of PIO on SREBP1 expression (**h**), Fillipin III staining (**i**), and percentages of IFN-γ⁺ iNKT cells and MFI of anti-IFN-γ (**j**) in TI iNKT cells from HCC patients. Data are means ± SEM of 13 patients (**a–c**), 6 patients (**f, h**), 5 patients (**g, i**), or 8 patients (**j**). Data were analyzed by paired student's *t*-test (**a–c**), Mann–Whitney test (**f–j**), or Pearson's correlation test (**d, e**). *P < 0.05, **P < 0.01, ***P < 0.001. Source data are provided as a Source Data file.

were positively correlated with the level of PPARγ in cells from tumors and para-carcinoma tissues (Fig. 5d, e). Furthermore, in line with the important roles of PPARγ and cholesterol in promoting mouse iNKT cell function (Fig. 2d, i, j, 3e, f), intratumoral human iNKT cells significantly reduced IFN-γ production, as indicated by frequencies of IFN-γ+ iNKT cells and mean fluorescence intensity of intracellular anti-IFN-γ (Fig. 5f). To demonstrate that the dysfunction of intratumoral iNKT cells was due to impaired cholesterol synthesis, we added water-soluble cholesterol to tumor-infiltrating iNKT cells and found that it restored IFN-γ production successfully (Fig. 5g). Consistently, PIO, the agonist of PPARγ, successfully restored SREBP1 expression, cholesterol synthesis, and IFN-γ production in human tumor-infiltrating iNKT cells (Fig. 5h-j). These results demonstrate that reduced PPARγ causes impaired cholesterol synthesis and dysfunction of human tumor-infiltrating iNKT cells, and that could be restored by PIO.

**PIO restores function of mouse intratumoral iNKT cells**. To further elucidate the role of PPARγ in regulating iNKT cell function in tumors, mice subcutaneously inoculated with B16F10 melanoma cells were used in our studies. In comparison with splenic iNKT cells, tumor infiltrating-iNKT cells reduced IFN-γ (Fig. 6a), PPARγ (Fig. 6b), SREBP1 (Fig. 6c), and cholesterol (Fig. 6d). These results were in line with those findings in human intratumoral iNKT cells (Fig. 5a–f). Tumor microenvironment is enriched with lactic acid, which has been shown previously to inhibit IFN-γ production from iNKT cells[35]. Here, we found that lactic acid significantly reduced PPARγ, and hence inhibited expression of SREBP1 and synthesis of cholesterol in iNKT cells (Supplementary Fig. 10a). In agreement with the previous findings[35], reduced mTOR protein and impaired activation of mTORC1 were detected in iNKT cells treated with lactic acid (Supplementary Fig. 10b). mTORC1 pathway has been reported to promote transcription of *Pparg* in CD4 T cells[25]. Using rapamycin to inhibit mTORC1 pathway, we showed that activation of mTORC1 was required for expression of PPARγ and hence for SREBP1 expression and cholesterol synthesis in iNKT cells (Supplementary Fig. 10c). These results indicate that reduction of PPARγ in tumor-infiltrating iNKT cells is caused by lactic acid in tumor microenvironment.

To test whether PIO could restore intratumoral iNKT cell function in vivo, B16F10 melanoma-bearing mice received PIO daily for 7 days via oral gavage, and then the mice were administered with iNKT cell-specific antigen αGC intraperitoneally (Fig. 6e). PIO elevated SREBP1 expression and cholesterol synthesis in tumor infiltrating-iNKT cells after activation (Fig. 6f, g). Consequently, PIO treatment significantly increased αGC-induced IFN-γ production in those intratumoral iNKT cells (Fig. 6h). Moreover, we confirmed that PIO treatment promoted synaptic TCR accumulation and synapse formation of tumor-infiltrating iNKT cells in vitro (Fig. 6i-k).

**PIO promotes iNKT cell-mediated anti-tumor immune responses**. It is well known that activation of iNKT cells by αGC or αGC-pulsed DCs inhibits tumor growth and provides survival benefit in mouse and in human patients[5–8]. The anti-tumor efficacy of iNKT cells is correlated to their IFN-γ production[3,36,37]. Although PIO did not completely restore the lipid synthesis and IFN-γ production in intratumoral iNKT cells (Fig. 6h), augmentation of IFN-γ in tumors indicated a better anti-tumor response. In agreement with the restoration of IFN-γ production in tumor-infiltrating iNKT cells (Fig. 6h), PIO treatment significantly promoted efficacy of iNKT cell-based immunotherapy against B16F10 melanoma cells (Fig. 7a-d). PIO or

αGC alone only modestly reduced tumor size and prolonged survival of tumor bearing mice, whereas combination of PIO and αGC did so strongly (Fig. 7b-d). As important immune regulatory cells, iNKT cells promote downstream activation of NK and CD8+ T cells indirectly[5,38,39]. Consistently, PIO and αGC in combination rather than PIO or αGC alone caused higher frequencies of intratumoral NK and CD8+ T cells (Fig. 7e), and induced more IFN-γ production from those cells (Fig. 7f). These results indicated that PIO plus αGC enhanced the activities of anti-tumor networks downstream iNKT cells. Conversely, in iNKT cell deficient *Jα18−/−* mice and in *Ifng−/−* mice, combination of PIO and αGC showed no influence on the tumor size, frequencies of intratumoral NK cells and CD8+ T cells, or their IFN-γ production (Fig. 7g-k). These results further confirmed that PIO plus αGC-augmented anti-tumor immune responses were dependent on iNKT cells and IFN-γ production. To demonstrate that PIO enhanced iNKT cell-mediated anti-tumor immune responses in vivo by targeting at PPARγ in iNKT cells, we generated mixed bone marrow chimeric mice to delete PPARγ specifically in iNKT cells. Mixed bone marrow cells from *Jα18−/−* mice and pLck-cre *Pparg fl/fl* mice (1:1 ratio) were injected intravenously into irradiated mice to create iNKT cell-specific deletion of PPARγ (Fig. 7l, m). As wide type controls, irradiated mice received mixed bone marrow cells from *Jα18−/−* mice and *Pparg fl/fl* mice (1:1 ratio). In line with previous findings that PLZF-cre *Pparg fl/fl* mice had normal numbers of peripheral iNKT cells, deficiency of PPARγ specifically in iNKT cells did not influence the intratumoral iNKT cell numbers in chimeras (Fig. 7n). However, in chimeras, deficiency of PPARγ in intratumoral iNKT cells reduced their IFN-γ production after PIO plus αGC treatment (Fig. 7o). Notably, chimeric mice with iNKT cell-specific deletion of PPARγ exhibited larger tumor size and lower survival than their wide type controls in response to PIO plus αGC treatments (Fig. 7p, q). These results further confirmed that PPARγ in iNKT cells was essential for the increased anti-tumor efficacy of PIO plus αGC treatment. In addition, combination of PIO and αGC enhanced efficacy of iNKT cell-based immunotherapy in MC38 colorectal cancer mouse model as well. PIO and αGC in combination but not alone significantly reduced tumor size and prolonged survival of MC38-bearing mice (Supplementary Fig. 11a, b). Taken together, PIO restores lipid synthesis and IFN-γ production in tumor-infiltrating iNKT cells and enhances the iNKT cell-mediated anti-tumor immune responses.

## Discussion

Lipid metabolism has been shown to regulate differentiation and functions of immune cells. Inhibition of ACC1, a key enzyme for de novo synthesis of fatty acids, favors development of regulatory T cells but inhibits Th17 differentiation[40]. Although expansion of CD8+ T cells depends on lipid synthesis, development of memory CD8+ T cells requires lipolysis to support their fatty acid β-oxidation[23,41]. Here, we report an important role of lipid synthesis, especially cholesterol synthesis, in promoting activation, proliferation, and effector function of iNKT cells. On the other hand, inhibition of fatty acid β-oxidation showed no influence on iNKT cell function (Supplementary Fig. 4). Although iNKT cells release both Th1 and Th2 cytokines upon activation, their IFN-γ production requires longer TCR signaling than IL-4 production[42,43]. Lack of cholesterol inhibited TCR signaling transduction (Fig. 3h-k), and thereby significantly diminished IFN-γ production whereas only showed minor effect on IL-4 (Fig. 3e). The polarization of iNKT cell function toward Th2 favors tumor growth rather than inhibits tumor growth. Therefore, alteration of lipid metabolism in iNKT cells would influence

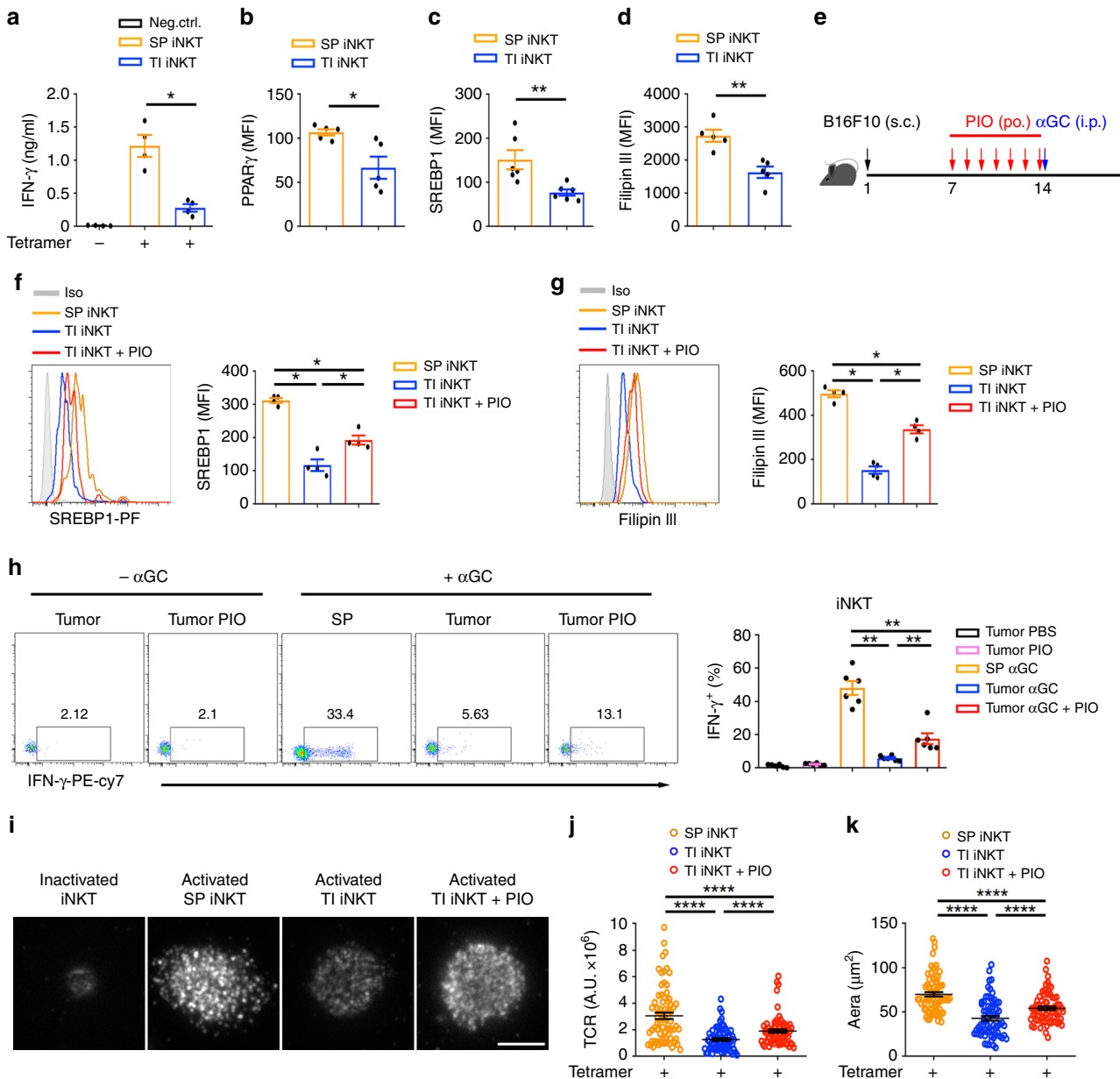

**Fig. 6 PIO restores cholesterol and IFN-γ production in tumor-infiltrating iNKT cells in vivo. a** IFN-γ production in tumor-infiltrating (TI) iNKT cells and splenic (SP) iNKT cells from B16F10 tumor-bearing mice, after activating with plate-coated mCD1d-PBS57 tetramer. **b–d** PPARγ (**b**), SREBP1 (**c**), and Filipin III staining (**d**) in TI iNKT cells and SP iNKT cells from B16F10 tumor-bearing mice. **e** Timeline of experimental procedure. **f–h** SREBP1 expression (**f**) and Filipin III staining (**g**) of iNKT cells, and percentages of IFN-γ+ iNKT cells (**h**) from indicated tissues of B16F10 tumor-bearing mice, after treating with or without PIO for 7 days and injecting with 2 μg αGC for 5 h. **i–k** Distribution (**i**) and fluorescence intensity (**j**) of surface TCR, and area of immunological synapse (**k**) of iNKT cells, after activating with coverslip-coated CD1d-PBS57 tetramer for 45 min. Cells from indicated tissues were isolated from mice treating with or without PIO for 7 days (n = 80 cells per group). Bar, 5 μm. Data are means ± SEM of four mice (**a**, **f**, **g**), five mice (**b**, **d**), or six mice (**c**, **h**), pooled from three independent experiments. Data were analyzed by Mann–Whitney test (**a–d**, **f–h**) or unpaired Student's t-test (**j**, **k**). *P < 0.05, **P < 0.01, ****P < 0.0001. Source data are provided as a Source Data file.

the progression of diseases. Notably, the roles of cholesterol in regulating T cell functions are not consistent[44,45]. Our results are in agreement with previous findings that cholesterol in cell membrane promotes TCR signaling in CD8+ T cells[30].

A recent study indicates that PPARγ promotes early T cell activation via enhancing fatty acid uptake[25]. However, PPARγ regulated biosynthesis rather than uptake of cholesterol in iNKT cells (Supplementary Fig. 7). It is well known that synthesis of cholesterol is controlled by SREBP1[46–48]. Interestingly,

transcription of *Srebf1* was regulated by PPARγ only in the presence of PLZF (Fig. 4n-p). In conventional T cells and adipose tissue iNKT cells which lack of PLZF expression, PPARγ failed to promote SREBP1 expression (Supplementary Fig. 8a-d, 9e). These results are in line with previous findings that TCR-mTOR signaling upregulates PPARγ but shows no influence on SREBP1 in CD4+ T cells[25]. The cell type-specific role of PPARγ is mediated by its cooperation with PLZF, a master transcriptional regulator in iNKT cells. Similar cell type-specific role of PPARγ has been

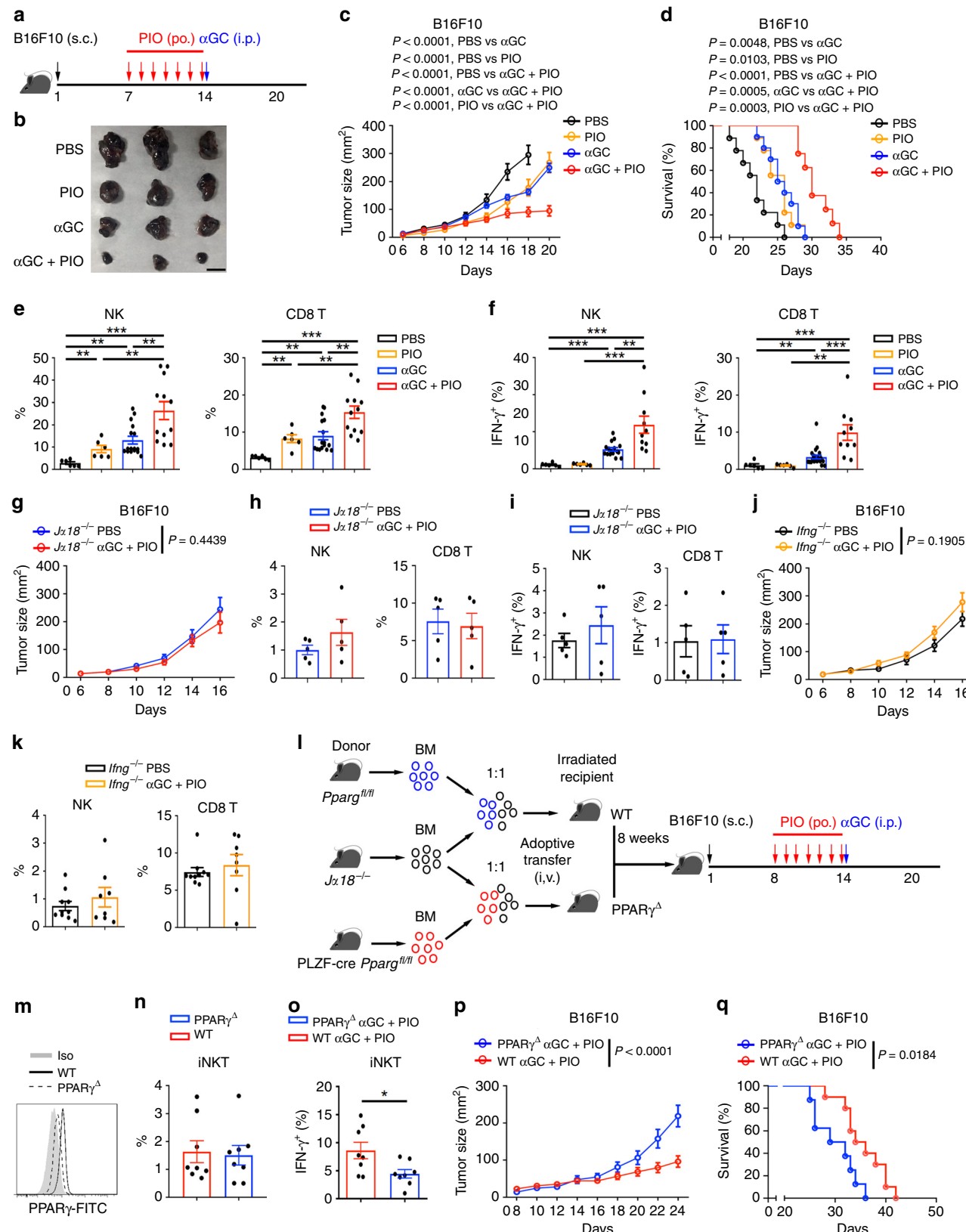

reported previously in adipose tissue Tregs via cooperation with Foxp3[49]. Notably, transcription of *Srebf1* was promoted by PPARγ2 rather than by PPARγ1 (Fig. 4n-q). These results were further explained by the findings that PPARγ2 but not PPARγ1 interacted with PLZF (Fig. 4r). Although it is still unclear whether PPARγ2 binds PLZF directly or indirectly, the extra 30 amino acids at N terminus of PPARγ2 are required for its interaction with PLZF. Moreover, we showed that PLZF promoted binding of PPARγ2 to *Srebf1* promoter region (Fig. 4s). Further studies are required to characterize the PPREs at *Srebf1* promoter region and to demonstrate the mechanisms by which PLZF interacts with PPARγ2 and promotes its binding to those PPREs. Our data

**Fig. 7 PIO promotes iNKT cell-mediated anti-tumor immunity. a** Timeline of experimental procedure. **b–d** Tumor pictures on day 20 (**b**), tumor size (**c**), and survival rate (**d**) of B16F10 tumor-bearing mice receiving indicated treatments (PBS, $n = 9$; PIO, $n = 9$; αGC, $n = 10$; αGC + PIO, $n = 8$). Bar, 1 cm. **e, f** Percentages of NK and CD8$^+$ T cells (**e**; PBS, $n = 6$; PIO, $n = 6$; αGC, $n = 16$; αGC + PIO, $n = 12$), and percentages of IFN-γ$^+$ NK and IFN-γ$^+$ CD8$^+$ T cells (**f**; PBS, $n = 6$; PIO, $n = 6$; αGC, $n = 16$; αGC + PIO, $n = 10$) in tumors from PIO-treated and PIO-untreated mice after injecting with or without 2 μg αGC for 5 h. **g** Size of tumors from $J\alpha18^{-/-}$ mice receiving indicated treatments (PBS, $n = 8$; αGC + PIO, $n = 7$). **h, i** Percentages of NK and CD8$^+$ T cells (**h**), and percentages of IFN-γ$^+$ NK and IFN-γ$^+$ CD8$^+$ T cells (**i**) in tumors from $J\alpha18^{-/-}$ mice receiving indicated treatments as described in **f** ($n = 5$ mice per group). **j** Size of tumors from $Ifng^{-/-}$ mice receiving indicated treatments (PBS, $n = 11$; αGC + PIO, $n = 9$). **k** Percentages of NK and CD8$^+$ T cells in tumors from $Ifng^{-/-}$ mice receiving indicated treatments (PBS, $n = 10$; αGC + PIO, $n = 8$). **l** Generation of mixed bone marrow chimeras. **m** PPARγ expression in iNKT cells from PPARγ$^\Delta$ mice or WT mice. **n** Percentages of intratumoral iNKT cells and percentages of IFN-γ$^+$ iNKT cells in tumors from PPARγ$^\Delta$ or WT mice receiving PIO daily and 2 μg αGC for 5 h ($n = 8$ mice per group). **p, q** Tumor size (**p**) and survival rate (**q**) of tumor-bearing PPARγ$^\Delta$ ($n = 8$) or WT ($n = 10$) mice receiving indicated treatments. Error bars represent SEM. Data were analyzed by two-way ANOVA (**c, g, j, p**), Mann–Whitney test (**e, f, h, i, k, n, o**), or log-rank test (**d, q**). *$P < 0.05$, **$P < 0.01$, ***$P < 0.001$. Source data are provided as a Source Data file.

could not exclude the possibility that other proteins may be involved. In contrast to the role of PPARγ in promoting Th1 response in iNKT cells, the functions of PPARγ in regulating conventional T cells are quite controversial. Although PPARγ does not regulate lipid synthesis in CD4$^+$ T cells, it has been reported to promote lipids uptake and cell proliferation in CD4$^+$ T cells[25]. In addition, it has been shown to promote differentiation and effector function of Th2 cells, whereas shows no influence on Th1 cells[50]. Conversely, suppressive effects of PPARγ on Th1 cytokine production[51,52], Th2 cytokine production, and T cell proliferation[53,54] are reported as well. The mechanisms underlying these discrepancies remain unclear, and cell type-specific regulators might be involved.

iNKT cells play important roles in tumor clearance. However, tumor microenvironment influences their cellular metabolism and hinders their anti-tumor functions. Reduction of PPARγ and IFN-γ was observed in tumor-infiltrating iNKT cells in both human patients and mouse models (Figs. 5a–f, 6a-d). Notably, impaired lipid synthesis is not the only reason dampening anti-tumor effects of iNKT cells. Low pH in microenvironment and impaired glycolysis would contribute to dysfunction of intratumoral iNKT cells as well[35,43]. Here, we reported that lactic acid downregulated PPARγ and inhibited cholesterol synthesis in iNKT cells (Supplementary Fig. 10a). Cholesterol only modulated proximal TCR signaling (Fig. 3g-k), whereas lactic acid inhibited IFN-γ production even TCR signaling was bypassed[35]. Therefore, PPARγ-cholesterol pathway is not the only pathway mediating suppressive effect of lactic acid. For these reasons, PIO only restored IFN-γ production in intratumoral iNKT cells to some extent in vivo (Fig. 6h). Nevertheless, it significantly promoted anti-tumor efficacy of iNKT cell-based immunotherapy in animal models (Fig. 7b-d). Here, PIO alone only slightly increased frequencies of intratumoral NK and CD8$^+$ T cells and showed no effect on their IFN-γ production (Fig. 7e, f). Significantly elevated NK and CD8$^+$ T cell numbers and their IFN-γ production were only observed in mice receiving both PIO and αGC (Fig. 7e, f). Moreover, in the absence of iNKT cells, combination of PIO and αGC failed to promote accumulation and activation of NK and CD8$^+$ T cells in tumors (Fig. 7g-i). Therefore, NK and CD8$^+$ T cell-mediated anti-tumor immune responses were promoted by iNKT cells. It has been reported that activation of iNKT cells indirectly promotes downstream activation of NK and CD8$^+$ T cell. After injecting αGC, iNKT cells release IFN-γ, activate DCs reciprocally, and induce IL-12 production from DCs. Sequentially, matured DCs activate NK and CD8$^+$ T cells, and IFN-γ and IL-12 further promote the activation of NK and CD8$^+$ T cells[2,38,55,56]. Therefore, activation of iNKT cells efficiently enhances multiple anti-tumor immune responses. These direct and indirect tumor-killing manners of iNKT cells make them ideal candidates for anti-tumor immunotherapies. In $Ifng^{-/-}$

mice, PIO and αGC in combination did not reduce tumor size or increase the intratumoral NK and CD8$^+$ T cells (Fig. 7j, k). These results further confirmed that IFN-γ was the key factor mediating iNKT cell-based immunotherapy against tumors. Our results could not exclude the possibility that PIO may also directly influence the function of NK and CD8$^+$ T cells when they are transactivated downstream iNKT cells. In addition, previous studies demonstrate a role of PPARγ in promoting CD1d and Cathepsin D expression in DCs[57,58]. Here, the contribution of PIO on lipid antigen presentation could not be excluded as well. However, with mixed bone marrow chimeric mice in which PPARγ was specifically deleted in iNKT cells, we confirmed that PPARγ expression in iNKT cells was required for the anti-tumor efficacy of PIO and αGC treatment (Fig. 7o-q). These results demonstrate that PIO targets PPARγ in iNKT cells and enhances their anti-tumor efficacy.

Together, we demonstrate that impaired lipid synthesis hinders anti-tumor effect of tumor-infiltrating iNKT cells. Restoring lipid synthesis via activating PPARγ recovers αGC-induced IFN-γ production and significantly enhances the efficacy of iNKT cell-based immunotherapy against tumors. Our results exhibit a promising prospect for combinational therapies targeting iNKT cells and their cellular metabolism. Importantly, PIO has been already used to treat type 2 diabetes, and that warrants its potential application for anti-tumor treatment in clinic.

## Methods

**Mice.** $V\alpha14$ Tg.$cxcr6^{gfp/+}$ mice were previously described[59] and provided by Dr. Albert Bendelac. All mice used were on the C57BL/6J background and were 8 to 12 weeks old. Cohoused littermates were used in our experiments. Mice were housed under specific pathogen-free conditions. All animal procedures were approved by the Animal Care and Use Committee of University of Science and Technology of China, and all experiments were performed in accordance with the approved guidelines.

**Clinical biospecimens.** Clinical tumor biospecimens were obtained from HCC patients in accordance with protocols approved by the Biomedical Ethics Committee of Anhui Medical University. Informed consent was obtained from all subjects.

**Cell isolation and activation.** CD4$^+$ T cells or CD8$^+$ T cells were enriched from spleens of wide type mice. Purified iNKT Cells, gating as GFP$^{hi}$ cells, were sorted from livers or spleens of $V\alpha14$ Tg.$cxcr6^{gfp/+}$ mice by FACS Aria (BD Biosciences), and were stimulated with plate-coated anti-CD3 (5 μg ml$^{-1}$) plus anti-CD28 (5 μg ml$^{-1}$) antibodies or with plate-coated mCD1d-PBS57 tetramer (0.5 μg ml$^{-1}$) overnight. In some experiments, iNKT cells were activated by PMA (50 ng ml$^{-1}$) plus ionomycin (1 μM) for 4 h. Cytokines in supernatants were measured by cytometric bead array kit (BD Biosciences). To measure the expression of activation markers, iNKT cells were stimulated with plate-coated anti-CD3 plus anti-CD28 antibodies overnight. In proliferation assays, cells were stimulated with plate-coated anti-CD3 plus anti-CD28 antibodies for 2 days, and cell proliferation was measured by Ki67 staining (BD Biosciences). To inhibit lipid synthesis in vitro, cells were treated with T007 at 2 μg ml$^{-1}$, with PIO at 10 μM, with GW9662 at 4 μg ml$^{-1}$, with Tofa at 10 μM, with C75 at 10 μM, and with simvastatin at 10 μM,

respectively. To study the influences of cholesterol on TCR signaling and iNKT cell function, water-soluble cholesterol was added to cells at 1 μM. To investigate the effects of GW9662 on iNKT cells in vivo, mice were administered intraperitoneally with 2 mg kg$^{-1}$ GW9662 or PBS buffer 20 h before injecting 2 μg αGC. Mice were sacrificed 4 h later, and intracellular IL-4 and IFN-γ in hepatic iNKT cells were measured by flow cytometry.

To isolate tumor-infiltrating lymphocytes from tumor-bearing mice or HCC patients, tumors were cut into small pieces and digested with collagenase IV (1 mg ml$^{-1}$) for 60 min at 37 °C, and followed by centrifugation in Percoll density gradient. Tumor-infiltrating iNKT cells from tumor-bearing mice were enriched with mCD1d-PBS57 tetramer, and were stimulated with plate-coated mCD1d-PBS57 tetramer for cytokine measurement. Tumor-infiltrating iNKT cells isolated from HCC patients were stimulated with plate-coated hCD1d-PBS57 tetramer (0.5 μg ml$^{-1}$) for 5 h in vitro, and intracellular IFNγ was measured by flow cytometry.

**Antibodies and flow cytometry**. Fluorochrome-labeled or unlabeled monoclonal antibodies against mouse TCRβ (Biolegend 109222, 1:200 dilution), CD4 (Biolegend 100408, 1:200 dilution), CD8α (Biolegend 100706, 1:200 dilution), CD69 (Biolegend 104522, 1:200 dilution), CD25 (Biolegend 102030, 1:200 dilution), CD3 (Biolegend 100331, 1:200 dilution), CD28 (Biolegend 102112, 1:200 dilution), NK1.1 (Biolegend 108706, 1:200 dilution), IFN-γ (Biolegend 505826, 1:200 dilution), IL-4 (Biolegend 504104, 1:200 dilution), PLZF (Biolegend 145804, 1:200 dilution), mTOR (Cell signaling Tech 2983S, 1:200 dilution), SREBP1 (Abcam ab3259, 1:100 dilution), PPARγ (Santa cruz sc-7273 FITC, 1:100 dilution), human TCRβ (Biolegend 306708, 1:200 dilution), and human IFN-γ (Biolegend 502528, 1:200 dilution) were used in our experiments. mCD1d-PBS57 tetramer and hCD1d-PBS57 tetramer were provided by the NIH Tetramer Core Facility. Intracellular lipids was stained with 0.5 μg ml$^{-1}$ BODIPY 493/503 (Invitrogen) for 20 min at 37 °C. Samples were acquired by a BD FACSVerse flow cytometry and data were analyzed with FlowJo software (TreeStar).

**Quantification of long-chain fatty acids**. iNKT cells (1 × 10$^7$) were sorted and stimulated with plate-coated anti-CD3 plus anti-CD28 for 24 h with or without T007, or PIO. Lipids were extracted to analyze fatty acid composition as described[26]. In brief, C-17 heptadecanoic acid was added to cell pellets as an internal standard for fatty acids. Cells were dried and lipids extracted using chloroform: methanol. The lipid extract was saponified and fatty acids were derivatized to methyl ester forms with methanolic boron trifluoride, extracted into hexane, and then injected into an Agilent 7890B/5977 gas chromatography-mass spectrometry (GC-MS) fitted with an Agilent DB-225 column.

**Western blot**. To expand iNKT cells in vitro, splenocytes from Vα14 Tg mice were cultured with 100 U ml$^{-1}$ human recombinant IL-2 and 100 ng ml$^{-1}$ αGC for 3 days, then were cultured for another 7 days with only 100 U ml$^{-1}$ human recombinant IL-2. Expanded iNKT cells were activated by immobilized anti-CD3 plus anti-CD28 for 20 min to measure the activation of TCR signaling pathways, and were activated overnight for measuring expression of SREBP1, SREBP2, and interaction between PLZF and PPARγ. Cells were harvested and lysed with sample buffer and were boiled for 15 min. Proteins were separated by electrophoresis and were detected by western blot. Antibodies against PPARγ (Santa cruz sc-7273, 1:1000 dilution), SREBP1 (Abcam ab3259, 1:1000 dilution), SREBP2 (Abcam ab30682, 1:1000 dilution), PLZF (Santa cruz sc-28319, 1:1000 dilution), P-LCK (Sigma SAB4300118, 1:1000 dilution), P-LAT (Cell signaling Tech 3584, 1:1000 dilution), LCK (Cell signaling Tech 2984, 1:1000 dilution), LAT (Cell signaling Tech 9166, 1:1000 dilution), and β-actin (Transgen Biotech HC201-02, 1:1000 dilution) were used in our experiments.

**RNA isolation and qPCR**. Total RNA was isolated with the ReliaPrep™ RNA Cell Miniprep System (Promega Z6010) and was reverse transcribed with Reverse Transcription System (Promega A3500). Real-time qPCR was performed with PikoReal 96 (Thermo Scientific). In each experiment, samples were run in triplicate and were normalized to β-actin to determine relative expression levels. Primer sequences were listed in Supplementary Methods.

**Knockdown of Pparg or Srebf1**. To knock down Pparg or Srebf1, synthesized DNA short-hairpin sequences targeting Pparg (5′-GGG TGA AAC TCT GGG AGA TTC-3′) or Srebf1 (5′-GCC TGC TAT GAG GAG GGT ATT-3′) were ligated into pGPU6/GFP/Neo vector. Scrambled shRNA (5′-TTC TCC GAA CGT GTC ACG T-3′) was used as negative control. ShRNA was introduced into cells by electroporation using a mouse T-cell nucleofector Kit (Lonza). After 36 h, cells were activated by plate-coated anti-CD3 plus anti-CD28 antibodies for 15 h. GFP$^+$ iNKT cells were analyzed and intracellular IFNγ was measured by flow cytometry.

**Measurement of the cholesterol**. Filipin III was dissolved in DMSO to reach the final concentration of 25 mg ml$^{-1}$. Cells were fixed with 4% paraformaldehyde (PFA) and stained with 50 μg ml$^{-1}$ filipin III for 60 min at 4 °C.

In some experiments, the total cellular cholesterol level was quantified using the Amplex Red cholesterol assay kit (Invitrogen). To quantify the intracellular cholesterol, cells were fixed with 0.1% glutaraldehyde and then were treated with 8 U ml$^{-1}$ cholesterol oxidase for 30 min to oxidize the plasma membrane cholesterol. The intracellular cholesterol was then quantified using the Amplex Red cholesterol assay kit. The value of the plasma membrane cholesterol was obtained by subtracting the intracellular cholesterol from the total cellular cholesterol.

**Total internal reflection fluorescence microscopy**. Eighteen millimeter round coverslips (Thermo) were cleaned and coated with poly-L-lysine for 15 min. mCD1d-PBS57 tetramer (0.5 μg ml$^{-1}$) or anti-CD3 (5 μg ml$^{-1}$) plus anti-CD28 (5 μg ml$^{-1}$) were used to coat the coverslips for 60 min at 37 °C. iNKT cells were seeded onto coverslips for 30 min with or without T007, Tofa, or simvastatin. To detect surface TCR distribution, cells were stained with Alexa-647-labeled anti-TCRβ (5 μg ml$^{-1}$) for 60 min on ice, and then were fixed in 4% paraformaldehyde for 15 min on ice. To view the location of PPARγ and PLZF, iNKT cells were fixed with 4% paraformaldehyde and permeabilized with buffer containing 5% FBS and 0.1% Triton X-100 for 30 min on ice, and then cells were stained with PE labeled anti-PLZF and FITC labeled anti-PPARγ for 60 min. Samples were visualized with a Leica SR GSD microscope at 100 nm penetration depth. Quantitative analyses were performed using ImageJ software.

**Chromatin immunoprecipitation (ChIP)-qPCR**. Expanded iNKT cells were activated by immobilized anti-CD3 plus anti-CD28 overnight in the presence or absence of T007. Cells were harvested and cross-linked with 1% formaldehyde at room temperature for 15 min, followed by neutralization with 125 mM glycine. Digestion and isolation of protein-bound DNA was performed using a magnetic ChIP kit (Thermo scientific). Antibody against PPARγ (Abcam ab41928, 1:100 dilution) and IgG control antibody (Santa Cruz sc-2025, 1:100 dilution) were used for ChIP. To detect the binding sites of PPARγ at srebp1 promoter region, different sets of primers covering the −2000 bp to +1 bp region of Srebf1 promoter were used for qPCR experiments, and were listed in Supplementary Methods. The fold enrichment is normalized to IgG control.

**Mouse tumor models**. In B16F10 melanoma models, B16F10 cells (1 × 10$^5$) were subcutaneously injected into the oxter part of mice. From day 6, tumor size was measured every 2 days, and animal survival rate was recorded every day. In MC38 models, MC38 cells (1 × 10$^6$) were subcutaneously injected into the oxter part of mice. From day 8, tumor size was measured every 2 days, and animal survival rate was recorded every day. Tumor size was calculated as length × width. Mice with tumor size larger than 20 mm at the longest axis were euthanized for ethical consideration. From day 6 or 8, tumor-bearing mice were randomly divided into four groups and were intragastrically administered with PBS buffer or with PIO daily at the dose of 30 mg kg$^{-1}$. On day 14, 2 μg αGC was injected intraperitoneally. To analyze the functions of intratumoral iNKT cells, NK cells, and CD8$^+$ T cells, mice were euthanized 5 h after αGC injection, and IFNγ production were measured by flow cytometry using cytokine secretion assay kit (Miltenyi Biotec). To visualize the melanoma, mice were euthanized on day 20.

**Mixed bone marrow chimeras**. pLck-cre Pparg$^{fl/fl}$ or Pparg$^{fl/fl}$ bone marrow cells were, respectively, mixed with Jα18$^{-/-}$ bone marrow cells at 1:1 ratio, and then were intravenously transferred into lethally irradiated recipient mice (10 Gy). After 8 weeks, B16F10 melanoma models were generated with these chimeras.

**Statistical analysis**. Statistical analyses were performed with paired or unpaired Student's t-test, Mann–Whitney test, Pearson's correlation test, two-way analysis of variance (ANOVA), or log-rank test. Particularly, the paired t-test is used for normally distributed parameters in two paired samples, unpaired Student's t-test is used for normally distributed parameters in two unpaired samples, Mann–Whitney test is used for non-normally distributed parameters in two unpaired samples, and log-rank test is used for survival analysis. All statistical analyses were performed using GraphPad Prism software. *$p < 0.05$, **$p < 0.01$, ***$p < 0.001$, and ****$p < 0.0001$ were considered statistically significant.

**Reporting summary**. Further information on research design is available in the Nature Research Reporting Summary linked to this article.

## Data availability
There is no restriction in the availability of materials described in the study. The source data for Figs. 1a-g, 2a–i, 2k, 3a-g, 3i-k, 4a-g, 4i, j, 4l-s, 5a-j, 6a-h, 6j, k, 7c-k, 7m-q, and Supplementary Figs. 2a-d, 3b-d, 3a, 4b, 5a, b, 6b-d, 7a, b, 8a-h, 9b-e, 10a-c, 11a, b are provided as a Source Data file.

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

## Acknowledgements

We thank NIH Tetramer Core Facility for providing us mCD1d-PBS57 and hCD1d-PBS57 tetramers. This work was supported by National Key R&D Program of China 2017YFA0505300, National Natural Science Foundation of China 91942310, 81771671, 91954122, and 91542203, and the Strategic Priority Research Program of the Chinese Academy of Sciences XDA12030208 to L.B. In addition, this work was supported by National Natural Science Foundation of China 81770588 and 81522009 to H.W. and was supported by the Fundamental Research Funds for the Central Universities to S.C.F.

## Author contributions

S.C.F. and L.B. conceived the idea and designed the experiments, S.C.F., K.X.H., C.X.T., H.S., C.W. Z., S.Y.B., J.W.L., Q.L.W., D.X., and T.Y. performed experiments, Z.X.S., Q.Q.D., X.J.Y., S.Z., G.L., R.B.Z., S.Z.D., Z.G.T., T.X.; and H.W. provided materials and discussed experiments, S.C.F., H.W., and L.B. wrote the paper.

## Competing interests

The authors declare no competing interests.
