## [Peer Review File · Nature Communications]

Reviewers' comments:

Reviewer #1, expert in NKT biology (Remarks to the Author):

In this manuscript, it is convincingly shown that PPAR γ activity in natural killer T cells (NKT cells) is required for increased lipid biosynthesis. In particular, it is shown that cholesterol is necessary for optimal production of IFN- γ following activation. Of particular interest, it is shown that tumor infiltrating NKT cell responses can be rescued by treatment with pioglitazone. Overall, the experiments are well-crafted and the findings should be of significant interest to the research and medical community.

Derek Sant'Angelo

Reviewer #2, expert in NKT and lipid metabolism (Remarks to the Author):

This manuscript has investigated the mechanisms that render iNKT cell dysfunctional in the tumor microenvironment. Although this issue has been investigated in a variety of other published studies, mechanisms are likely multifactorial and complex. In this manuscript, a pathway is proposed involving induction by lactic acid in the tumor microenvironment to reduce mTORC1 activity, which inhibits PPAR γ production, and in turn reduces SREBP1 transcription factor activity and production of cholesterol to inhibit IFN- γ production by iNKT cells. Mechanistically, it is further shown that PPAR γ interacts with the innate transcription factor PLZF. These defects could be overcome with agonists of PPAR γ or by supplementation of cholesterol. Importantly, a PPAR γ antagonist was able to promote the antitumor activities of glycolipid α -GalCer-activated iNKT cells in vivo in mice. From these studies it is concluded that restoration of lipid biosynthesis may enhance the anti-tumor activities of iNKT cells in the clinical setting.

General comments:

The authors employ a variety of model systems, including knockout animals, human samples, siRNA, chemical modulators of protein activity, and lipids. A compelling and extensive set of experiments is performed in support of the findings.

Specific comments:

1. It is concluded that PLZF and PPAR γ 2 interact. Although not specifically noted, it is unlikely the authors can distinguish between direct versus indirect physical interactions at this time. Can they make a comment about this?
2. In their mouse tumor studies, the effects of PIO may be contributed, at least in part, by direct effects on cells other than iNKT cells; e.g., NK cells or CD8 T cells activated in trans. Although the studies with Ja18 knockout mice demonstrate specificity, they cannot exclude direct effects on NK and CD8 T cells. A small comment might be appropriate.
3. In the context of point 2, a previous study showed that PPAR γ can influence CD1d expression in dendritic cells, which could be briefly discussed in the context of the present findings.
4. The text uses the term "prove" in a few places, which could be replaced by "demonstrate," as biological experiments rarely can absolutely prove something is true or false.
5. Although the paper is written well, grammar and spelling could be improved in a few places.

Reviewer #3, expert in cancer immunology (Remarks to the Author):

iNKT cells carry potential for anti-tumor efficacy, but several studies have demonstrated functional defects in tumor-infiltrating iNKT cells. In this paper the authors suggest that PPAR γ impacts iNKT function through enhancing lipid biosynthesis and particularly cholesterol through enhanced transcription of *srebf1*. They show that *pparg* agonist pioglitazone led to enhanced iNKT function and synergized with Galcer for anti-tumor effects in murine models.

Overall, I think this is a nice paper that contains well executed and mechanistic experiments. The findings should be of interest to develop strategies to enhance iNKT based immunotherapies. I do believe the authors should address the following issues.

1. A major issue that applies throughout the paper is potential use of inappropriate statistical tests. I don't think that use of student T test is suitable with these sample sizes, and expected data distribution. Unfortunately, this impacts majority of the panels in several figures, and may impact interpretation of data.

2. The concept that PPAR γ can impact T cell function is not new. More relevant here is which effects / mechanisms are cell-type specific for NKT cells. Drugs such as pioglitazone can have effects on multiple cell types in vivo and also have off target effects. Therefore the strong reliance on drug/small molecule effects throughout the paper may suggest the need for caution. In this regard, there are some key experiments such as Fig2K and suppl fig 3 wherein mice have *pparg* deficient NKT cells. However these mice in turn seem to have decline in NKT numbers (suppl Fig 3). While this was "significant" only in the thymus, lack of "significance" in other tissues may simply be due to small numbers of mice studied. This may also impact some data interpretation, depending on experimental details. For example, unless equal numbers of cells were sorted in Fig 2K, apparent decline in IFN γ production may simply be lower proportion of starting iNKT cells.

3. An interesting and novel finding in this paper relates to direct interaction between *pparg* and *plzf*, which may be mechanistically important to cell-type specific effects. However this is mostly studied in 293T cells, and whether this interaction is important to enhance the expression of *srebp1* would clearly enhance the paper.

4. In terms of experiments with tumor-bearing mice, it would be important to more directly show/test whether *pparg* in NKT cells is essential for function of tumor-infiltrating NKT cells. This is again because pioglitazone could impact other cells, and targets as mentioned earlier. In fact, the data shown suggest that the effects are systemic and not specific for tumor-associated NKT cells.

Reviewers' comments:

Reviewer #1, expert in NKT biology (Remarks to the Author):

In this manuscript, it is convincingly shown that PPAR γ activity in natural killer T cells (NKT cells) is required for increased lipid biosynthesis. In particular, it is shown that cholesterol is necessary for optimal production of IFN- γ following activation. Of particular interest, it is shown that tumor infiltrating NKT cell responses can be rescued by treatment with pioglitazone. Overall, the experiments are well-crafted and the findings should be of significant interest to the research and medical community.

Response: We appreciate the reviewer's comments.

Reviewer #2, expert in NKT and lipid metabolism (Remarks to the Author):

This manuscript has investigated the mechanisms that render iNKT cell dysfunctional in the tumor microenvironment. Although this issue has been investigated in a variety of other published studies, mechanisms are likely multifactorial and complex. In this manuscript, a pathway is proposed involving induction by lactic acid in the tumor microenvironment to reduce mTORC1 activity, which inhibits PPAR γ production, and in turn reduces SREBP1 transcription factor activity and production of cholesterol to inhibit IFN- γ production by iNKT cells. Mechanistically, it is further shown that PPAR γ interacts with the innate transcription factor PLZF. These defects could be overcome with agonists of PPAR γ or by supplementation of cholesterol. Importantly, a PPAR γ antagonist was able to promote the antitumor activities of glycolipid α -GalCer-activated iNKT cells in vivo in mice. From these studies it is concluded that restoration of lipid biosynthesis may enhance the anti-tumor activities of iNKT cells in the clinical setting.

General comments:

The authors employ a variety of model systems, including knockout animals, human samples, siRNA, chemical modulators of protein activity, and lipids. A compelling and extensive set of experiments is performed in support of the findings.

Specific comments:

1. It is concluded that PLZF and PPAR γ 2 interact. Although not specifically noted, it is unlikely the authors can distinguish between direct versus indirect physical interactions at this time. Can they make a comment about this?

Response: Thanks for the suggestion. We discuss this issue in discussion section.

Page 18:

Although it is still unclear whether PPAR γ 2 binds PLZF directly or indirectly, the extra 30 amino acids at N terminus of PPAR γ 2 are required for its interaction with

PLZF. Moreover, we showed that PLZF promoted binding of PPAR γ 2 to *Srebf1* promoter region (Fig. 4r). Further studies are required to characterize the PPREs at *Srebf1* promoter region and demonstrated the mechanisms by which PLZF interacts with PPAR γ 2 and promotes its binding to those PPREs. Our data could not exclude the possibility that other proteins may be involved.

2. In their mouse tumor studies, the effects of PIO may be contributed, at least in part, by direct effects on cells other than iNKT cells; e.g., NK cells or CD8 T cells activated in trans. Although the studies with Ja18 knockout mice demonstrate specificity, they cannot exclude direct effects on NK and CD8 T cells. A small comment might be appropriate.

Response: Thanks for the suggestion. We now discuss this possibility in discussion section.

Page 20:

Our results could not exclude the possibility that PIO may also directly influence the function of NK and CD8⁺ T cells when they are transactivated downstream iNKT cells.

3. In the context of point 2, a previous study showed that PPAR γ can influence CD1d expression in dendritic cells, which could be briefly discussed in the context of the present findings.

Response: We appreciate the reviewer's suggestion. The possible influence of PPAR γ on antigen presentation is now discussed in discussion section.

Page 20:

Additionally, previous studies demonstrate a role of PPAR γ in promoting CD1d and Cathepsin D expression in DCs. Here, the contribution of PIO on lipid antigen presentation could not be excluded as well. On the other hand, with mixed bone marrow chimeric mice in which PPAR γ was specifically deleted in iNKT cells, we confirmed that PPAR γ expression in iNKT cells was required for the anti-tumor efficacy of PIO and α GC treatment (Fig. 7o-q).

4. The text uses the term "prove" in a few places, which could be replaced by "demonstrate," as biological experiments rarely can absolutely prove something is true or false.

Response: We appreciate the reviewer's comment. "prove" has been replaced by "demonstrate" in the manuscript.

5. Although the paper is written well, grammar and spelling could be improved in a few places.

Response: We appreciate the reviewer's comment. The paper has been carefully revised to improve the grammar.

Reviewer #3, expert in cancer immunology (Remarks to the Author):

iNKT cells carry potential for anti-tumor efficacy, but several studies have demonstrated functional defects in tumor-infiltrating iNKT cells. In this paper the authors suggest that PPAR γ impacts iNKT function through enhancing lipid biosynthesis and particularly cholesterol through enhanced transcription of *srebf1*. They show that *pparg* agonist pioglitazone led to enhanced iNKT function and synergized with Galcer for anti-tumor effects in murine models.

Overall, I think this is a nice paper that contains well executed and mechanistic experiments. The findings should be of interest to develop strategies to enhance iNKT based immunotherapies. I do believe the authors should address the following issues.

1. A major issue that applies throughout the paper is potential use of inappropriate statistical tests. I don't think that use of student T test is suitable with these sample sizes, and expected data distribution. Unfortunately, this impacts majority of the panels in several figures, and may impact interpretation of data.

Response: Thanks for bringing up this issue. We have used Mann-Whitney test in some experiments, including Fig.1 (a-c, f, g), Fig.2 (f-i, k), Fig.3 (a, d, g), Fig.4 (c-g, i, m, o, p, q, s), Fig.5 (f-j), Fig.6 (a-d, f-h), Fig.7 (e, f, h, i, k, n, o), Supplementary Fig. 2d, Supplementary Fig. 3 (b-d), Supplementary Fig. 4 (a, b), Supplementary Fig. 6 (b-d), Supplementary Fig. 7 (a, b), Supplementary Fig. 8 (b, d-h), Supplementary Fig. 9 (d, e), and Supplementary Fig. 10 (a, b), according to the data distribution.

The concept that PPAR γ can impact T cell function is not new. More relevant here is which effects / mechanisms are cell-type specific for NKT cells. Drugs such as pioglitazone can have effects on multiple cell types in vivo and also have off target effects. Therefore the strong reliance on drug/small molecule effects throughout the paper may suggest the need for caution. In this regard, there are some key experiments such as Fig2K and suppl fig 3 wherein mice have *pparg* deficient NKT cells. However these mice in turn seem to have decline in NKT numbers (suppl Fig 3). While this was "significant" only in the thymus, lack of "significance" in other tissues may simply be due to small numbers of mice studied. This may also impact some data interpretation, depending on experimental details. For example, unless equal numbers of cells were sorted in Fig 2K, apparent decline in IFN γ production may simply be lower proportion of starting iNKT cells.

Response: Thanks for raising this issue. We now address this question with both in vitro experiments (Fig. 2k) and in vivo experiments (Fig. 7l-q). Increased numbers of mice are now shown in Fig.2k, Fig.4f, g, and Suppl. Fig. 3b-d. Deleting PPAR γ in iNKT cells in PLZF-cre *Pparg*^{*fl/fl*} mice did not influence the iNKT cell numbers in spleens and livers (Suppl. Fig. 3c, d), but reduced their IFN- γ production in vitro (Fig.2k). In Fig. 2k, we measured the intracellular IFN- γ in each iNKT cells by flow

cytometry, thereby the results were not affected by cell numbers. Additionally, we showed that in PPAR γ deficient iNKT cells drugs showed no influence on their IFN- γ production in vitro. These results confirmed that PIO and T007 regulated IFN- γ production in iNKT cells by targeting PPAR γ in vitro. We understand the reviewer's concern on in vivo effect of PIO. To further confirm that PIO enhanced anti-tumor efficacy in vivo by targeting PPAR γ in iNKT cells, we generated mixed bone marrow chimeric mice in which PPAR γ was specifically deleted in iNKT cells (Fig. 7l). In response to PIO and α GC administration, PPAR γ deficient iNKT cells produced less IFN- γ , mice with iNKT cell-specific deletion of PPAR γ showed larger tumor size and lower survival (Fig. 7m-q). All these results further confirmed that PIO enhanced iNKT cell-mediated anti-tumor efficacy in vivo by targeting PPAR γ in iNKT cells.

Fig. 2k

Fig. 4f,g

Suppl. Fig. 3b-d

Fig. 7l

Fig. 7m-q

2. An interesting and novel finding in this paper relates to direct interaction between *pparg* and *plzf*, which may be mechanistically important to cell-type specific effects. However this is mostly studied in 293T cells, and whether this interaction is important to enhance the expression of *sreb1* would clearly enhance the paper.

Response: Thanks for the suggestion. We now show that co-overexpression of *PPARγ2* and *PLZF* could increase *SREBP1* expression in both 293T cells (Fig. 4o) and iNKT cells (Fig. 4q). The interaction between *PPARγ* and *PLZF* were shown in both iNKT cells (Fig. 4j-l) and 293T cells (Fig. 4r). When we overexpressed mouse *PPARγ1* and *PPARγ2* respectively in 293T cells, with or without co-overexpressing mouse *PLZF*, we found that overexpressing *PLZF*, *PPARγ1* or *PPARγ2* alone failed to promote the transcription and expression of *SREBP1* in 293T cells.

Co-overexpression of *PPARγ1* and *PLZF* also failed to do so. Interestingly, in 293T cells co-overexpressing *PPARγ2* and *PLZF*, we detected significantly increased amount of mature *SREBP1*, total *SREBP1* protein, and *Sreb1* mRNA (Fig. 4n-p). These results demonstrated that *PPARγ2* cooperated with *PLZF* to promote the transcription and expression of *SREBP1*. Consistently, co-overexpressing *PPARγ2* and *PLZF* in iNKT cells increased *SREBP1* expression as well, in comparison to co-overexpressing *PPARγ1* and *PLZF* (Fig. 4q). Furthermore, we demonstrated that only *PPARγ2* but not *PPARγ1* interacted with *PLZF*, and *PLZF* significantly enhanced binding of *PPARγ2* to the *Sreb1* promoter region (Fig. 4s).

Fig. 4j-s

3. In terms of experiments with tumor-bearing mice, it would be important to more directly show/test whether *pparg* in NKT cells is essential for function of tumor-infiltrating NKT cells. This is again because pioglitazone could impact other cells, and targets as mentioned earlier. In fact, the data shown suggest that the effects are systemic and not specific for tumor-associated NKT cells.

Response: Thanks for the raising this issue and we absolutely agree with the reviewer's opinion. To demonstrate that PIO enhanced iNKT cell-mediated anti-tumor immune responses *in vivo* by targeting at PPAR γ in iNKT cells, we generated mixed bone marrow chimeric mice to delete PPAR γ specifically in iNKT cells. Mixed bone marrow cells from *J α 18^{-/-}* mice and pLck-cre *Pparg^{fl/fl}* mice (1:1 ratio) were injected intravenously into irradiated mice to create iNKT cell-specific deletion of PPAR γ (Fig. 7l, m). As wide type controls, irradiated mice received

mixed bone marrow cells from *Jα18^{-/-}* mice and *Pparg^{fl/fl}* mice (1:1 ratio). In line with previous findings that PLZF-cre *Pparg^{fl/fl}* mice had normal numbers of peripheral iNKT cells, deficiency of PPAR γ specifically in iNKT cells did not influence the intratumoral iNKT cell numbers in chimeras (Fig. 7n). However, in chimeras, deficiency of PPAR γ in intratumoral iNKT cells reduced their IFN- γ production after PIO plus α GC treatment (Fig. 7o). Notably, chimeric mice with iNKT cell-specific deletion of PPAR γ exhibited larger tumor size and lower survival than their wide type controls in response to PIO plus α GC treatments (Fig. 7p, q). These results further confirmed that PPAR γ in iNKT cells was essential for the increased anti-tumor efficacy of PIO plus α GC treatment.

Fig. 7l-q